# The Guidance and Control of Urban Planning for Reuse of Industrial Heritage: A Study of Nanjing

Yanming Wu [1,2,*], Uta Pottgiesser [2,3], Wido Quist [2] and Qi Zhou [1]

1 Institute of Architectural History and Theory, School of Architecture, Southeast University, Nanjing 210096, China; 101001781@seu.edu.cn
2 Department of Architectural Engineering and Technology, Faculty of Architecture and the Built Environment, Delft University of Technology, 2628 BL Delft, The Netherlands; u.pottgiesser@tudelft.nl (U.P.); w.j.quist@tudelft.nl (W.Q.)
3 Detmold School of Architecture and Interior Architecture, Technische Hochschule Ostwestfalen-Lippe, University of Applied Sciences and Arts, 32756 Detmold, Germany
* Correspondence: y.wu-8@tudelft.nl; Tel.: +86-18934599967

**Abstract:** Industrial heritage is among the products of modern urban development, and the influencing factors of its regeneration and development are often complex. Due to different national conditions, the research progress and approaches in industrial heritage reuse in China are different from other countries. While industrial heritage sites in Europe have become part of urban redevelopment in several regions, China still focuses mostly on single objects, lacking systematic analysis, especially at the urban scale. Regarding the city of Nanjing, an operational approach to complex urban dynamics is proposed based on a simplified analysis of official statistics, maps and GIS technology. The influence mechanisms of Nanjing's urban planning on industrial heritage regeneration and development after 1990 are analyzed. The results show that urban growth boundaries, traffic accessibility, eco-environmental policies, population distribution, industrial renovation investment and natural resource change all have a significant impact on the abandonment and regeneration of Nanjing industrial heritage. This study expands the research perspective of industrial heritage reuse in China and proposes a clearer systematic planning strategy for the future of industrial heritage in cities.

**Keywords:** urban development; mechanism of influence; industrial heritage; reuse

## 1. Introduction

China started to implement the economic policy of reform and opening up after the Third Plenary Session of the Eleventh CPC (The Communist Party of China) Central Committee in December 1978. After initial attempts and setbacks, China's economic development accelerated again in the early 1990s on the premise of adhering to the reform and opening up policy [1]. Relying on the cost advantages of land and labor as well as institutional dividends, China quickly became a global manufacturing factory [2–4]. Against the backdrop of globalization, marketization and decentralization, the economic zones in the lower reaches of the Yangtze River, represented by Nanjing and other cities, have developed rapidly. Under the reform of a series of policies related to land, public-owned enterprises and taxation, the productive force of the original industrial enterprises in China's big cities has been liberated and rapidly improved [5,6].

With the advent of post-industrialism, manufacturing has experienced a process of suburbanization in major cities around the world. Many cities in Western countries began to suburbanize at the end of the 19th century, while some Chinese cities began to do so at the end of the 20th century [7–12]. Unfortunately, the rapid and unsustainable development mode in the past puts enormous pressure on the current development of some cities (e.g., Nanjing) in China based on land, space, environment, population, energy, resource,

etc. Due to the better service system and greater development potential, the historical urban areas in these cities, as the center of various modern urban functions, are still the focus of urban development and construction. The over-exploitation of these areas has further increased the burden and pressure on the service function and population of the city center [13]. In this context, in order to reduce the service pressure faced by the city center, many industrial buildings in the old town have been demolished or reused (Figure 1), and large areas of industrial land have been rearranged and put into use. These sites are of great value as cultural and historical relics and as material resources, and they can also play an important role in economic production through their reuse and repurposing [14]. However, due to various problems in the post-industrial era, efforts to protect and reuse industrial heritage in China have not been very successful in the past 30 years.

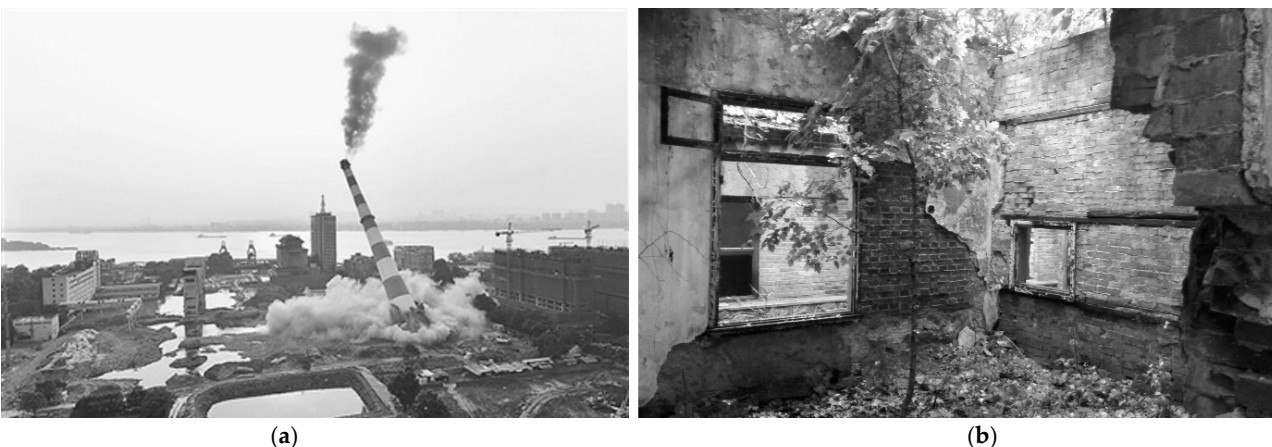

(**a**)　　　　　　　　　　　　　　　　　　　　　　　(**b**)

**Figure 1.** (**a**) The iconic chimney of the former National Government Capital Power Plant was demolished by blasting (Source: [15]); (**b**) The wall relics of Jiangnan Cement Factory built in the period of Republic of China (Source: [15]).

Specific reuse issues regarding China's post-industrialization are as follows:

1. In the process of urban development, the preservation of industrial heritage sites is influenced by many aspects, such as humanities, material, society, and politics, and these aspects involve many stakeholders: citizens, governments, developers and other groups. In the complex process of pursuing interests, some industrial heritage sites give way to urban development, while many old factories are idle due to the difficulty of reaching an agreement about the reuse purposes of industrial sites [16].
2. The original road system in industrial land usually has a special service function, which can only meet the needs of industrial production and transportation but cannot directly undertake new urban functions. When traffic and streamlining in these areas cannot effectively connect with the urban traffic system, these historical industrial areas may lose their significance and potential for reuse [17].
3. With urban development and expansion, the historic industrial areas in many cities have become the center of the city, and their land value has rapidly increased. However, due to the scattered layout and large area occupancy, the existence of old industrial land reduces the function agglomeration effect of urban central areas to some extent.
4. Sometimes, large old factories are not conducive to shaping the overall landscape of the city.
5. With the adjustment of the industrial layout and upgrading of the industrial structure, a large number of workers have to be laid off or are forced to retire. The huge amount of laid-off workers in old industrial areas is not conductive to the stability and harmony of urban society. At the same time, most of the residents in industrial areas usually belong to the low-income class in the city. The improvement of the overall

economic vitality of the city will face difficulty due to their limited consumption level [18].

6.  Most industrial enterprises will cause a certain degree of pollution to the urban eco-environment, and they also have a certain negative impact on the lives of urban residents. In addition, although the factories located in some industrial sites have stopped production, the damage caused to the urban eco-environment in the past has not been fully recovered [19].

Therefore, it is an unavoidable challenge for China's urban development to identify and implement strategies to deal with these industrial sites more rationally, ecologically and sustainably.

As early as the 1980s, Western countries began to pay attention to the redevelopment and utilization of abandoned industrial areas, which is usually included in the concept of brownfield regeneration [20]. Brownfields have been defined differently to serve different research purposes depending on the concerns of national governments, institutions, or interest groups. For example, in America, brownfields usually refer to: "Real property, the expansion, redevelopment, or reuse of which may be complicated by the presence or potential presence of a hazardous substance, pollutant, or contaminant" [21].

The cause of brownfield was that the change in the urban land value resulted in the recession of industrial areas and the adjustment of the urban industrial structure. Many cities no longer regard brownfield as a problem but have begun to realize the advantages of redeveloping such plots [22,23]. After treatment, brownfield can be developed into land for various purposes, such as retail business districts, residential areas, light pollution-free industrial areas, office areas, transportation hub stations, parks, squares, exhibition halls, and schools. Its aim is to recycle the land, so as to achieve the sustainable utilization of urban land. From the long-term perspective of urban development, the redevelopment of brownfield has many benefits, such as alleviating environmental pressure [24,25], stimulating economic growth [26], promoting social inclusion [27], and protecting local cultural [28]. Under the concept of sustainable development, today, the redevelopment of "brown land" has become one of the most important strategies for urban development in Western countries.

However, for many traditional industrial cities, the redevelopment of brownfield has to first consider the protection of a large number of industrial heritages. In the West, the initial focus on the potential value of industrial heritage arose in the field of industrial archaeology in late 19th century Britain, concentrating on the archaeological study of industrial remains originating from the period of the Industrial Revolution [29]. In the following years, this new independent discipline was widely accepted and spread to other European countries. Its concepts and methodology have thus been broadened considerably and, subsequently, industrial heritage has attracted attention at an international scale since the 1970s. In 1978, TICCIH (the International Committee for the Conservation of the Industrial Heritage), the first international society dedicated to the identification, evaluation and management of industrial heritage, was founded and became ICOMOS's specialist adviser to assess industrial heritage sites for the World Heritage List in 2000. From industrial archaeology to industrial heritage, and now to the industrial landscape, Western countries' attention to the value of industrial heritage has undergone an evolutional process. Generally speaking, industrial heritage has turned from specific interest in monuments (the individual building or a single machine) to industrial sites (including the machines, buildings, and their infrastructure), and then to whole industrial areas and industrial landscapes [30].

The Burra Charter declares that "Places may have a range of values for different individuals or groups" [31]. Values change over time because society is constantly in the process of revising what it values [32]. In general, the practice of defining Chinese industrial heritage shows a slight difference in understanding values and meanings regarding its specific social context. Taking its own industrial history as a reference, China has a distinctive scenario in terms of the juxtaposition of industrialization and deindustrialization [33]. Chinese early modern urbanization was marked by its industrial development during the

period of 1860–1937, and thus, the historical trajectory of industrial civilization could be seen as an integral part of the urban development process in China [34]. In this context, Shan noted that current studies on industrial heritage in China were mainly concerned with modern industrial history, starting from the second half of the 19th century, rather than dating back to the beginning of the Industrial Revolution before the 1860s [35]. In addition to the temporal factor, the categorization and geographical variation of China's industrialization in different regions and cities have led to tension in the definition of industrial heritage [36]. For example, the pace of industrialization in China's coastal areas was faster than that in inner-land areas, whilst the industrial development level in the plain and basin regions was higher than that in the mountain and plateau regions [37]. At present, we have seen Chinese industrial heritage extended to a diversity of buildings and objects considering industrial style, spatial distribution and temporal period [38].

Scholars in Western countries have conducted significant research on the reuse of industrial heritage by focusing on three areas. First, the causes and conditions are analyzed. Most of these studies attribute reuse motivation to visible urban problems, such as the decline in the manufacturing industry, the low rent of old industrial buildings, the low cost of reuse, the government's policy incentives, and the rise of the cultural tourism economy [39–42]. The second key research area is the means of reuse. The research perspectives, specific technical means and development strategies of industrial building preservation and reuse are the focus of architectural ontology research. Many aspects related to industrial heritage reuse have been analyzed in this area, such as building features, location features, policies and regulations, market target, and methods of public participation [43–49]. Finally, research focuses on the impacts on cities. Due to the diversity of the urban context, target setting and design methods, the impact of industrial heritage regeneration on buildings, surrounding blocks and the cities as a whole are complex and have already been studied in depth [30,50,51].

The existing research on the reuse of industrial heritage in China mainly concentrates on singular buildings from the perspective of the architecture discipline. According to the focus of research, they can be divided into the following four aspects. First is the design strategies in the process of reuse. This kind of research focuses on the specific design strategies carried out in the reuse practices of historical industrial buildings, aiming to classify the design involved in the process [52–54]. The second key research area is the relationship between conservation and reuse. The conservation concepts and methods applied to industrial heritage are very different compared to those applied to other general cultural relics. On the premise of preserving historical information as much as possible, these historical industrial buildings usually need to be injected into new functions. This means that there should be no "conflict" relationship between the two disposal strategies of protection and reuse, but "adaptability" should be achieved with appropriate reconcilement [55–58]. Third, there is reuse and urban regeneration. In this area, the positive roles of modern industrial heritage reuse are expounded, such as promoting the development of urban cultural industries, creating the spirit of urban places and shaping a new image for cities [59–61]. The final research area is case studies of reuse practice. In China, specific practical cases of protection and reuse have been extensively discussed and analyzed.

However, reuse is not an independent concept but a unified concept. The purpose of reuse may come from social, educational, research or innovation needs [62,63]. After a brief review of studies on industrial site reuse in China, it is not difficult to find that most studies are based on single or multiple cases. Although China's research regarding reuse has covered a large amount of industrial heritage sites, the number of comprehensive studies at the regional or city level is still quite insufficient. Moreover, the perspective of existing research on the reuse of industrial heritage mostly concentrates on the conservation strategy, evaluation method and significance, especially practical research, which is "practice-oriented". However, the complex motivation of regeneration, impacts of diverse urban contexts on reuse, and characteristics of spatial evolution processes lack systematic analysis. In addition, although the preservation concept of industrial heritage in China has

gradually developed from "case protection" to the international standard of "overall preservation", the research progress of industrial heritage groups on a regional or urban scale still lags behind. Thus, when faced with the contradiction between urban development and old industrial buildings, "overall preservation" is a difficult concept to implement in China.

The literature, in particular, some practical cases of reuse, shows that the reuse process is not only restricted by the characteristics of the building itself but is also regulated by a series of other factors [64]. On the urban scale, some scholars have conducted in-depth analyses on the impact of single factors and found that the land policy mainly influences the regeneration process of industrial heritage through the property right system [65]. Other related studies show that land policy and urban planning both have a significant impact on the reuse process of industrial projects [66]. In the study of historical cities, Spatial Syntax is used to analyze the spatial structure of cities and the causes behind them [67]. Even more, the reuse project must be harmonized with ethical principles and moreover draw elements from the urban planning and use of those products [68]. On the building scale, interdisciplinary methods are used to evaluate the preservation status of buildings and the influencing factors behind them [69,70]. Recently, some scholars have tried to explore the systematic mechanism on the influencing factors of industrial heritage reuse [41]. In addition, some studies discuss how culture affects the spread of protection technology and the influencing factors behind it [71]. According to the literature, six influencing factors have been identified: urban growth boundary, traffic accessibility, eco-environmental policies, population distribution, industrial renovation investment and natural resources.

Considering the existing relevant research, there is a focus on analyzing the influencing factors of industrial heritage reuse processes at the scale of buildings or blocks, and most studies select a single building or site as their main research object. However, at present, the research on the reuse of industrial heritage in traditional industrial cities still lacks systematic analysis on the urban scale, and current quantitative analyses on the influencing factors of reuse practice also seem to be insufficient.

To better establish and sharpen the focus of this paper, a main research question has been formulated after a thorough literature study of the problem field and background information involving the field of research. The main research question of this paper is as follows: How can the processes of industrial heritage abandonment and regeneration in Nanjing be better understood, and appropriately guided and controlled, through the close examination of the aspects influencing it on an urban scale? We hope to fill the gap in the existing research by answering this research question to some extent.

In order to answer this question, this paper analyzes the influence of urban planning on the processes of industrial heritage site abandonment and regeneration quantitatively, and it explores the comprehensive influence mechanism of urban planning in Nanjing. By using GIS technology, this study exhibits the processes of abandonment and regeneration of industrial heritage sites in Nanjing since 1990 and discusses the influence of different factors on this process. Finally, feasible strategies are proposed on how to protect and reuse industrial heritage sites as part of continuous urban development.

## 2. Materials and Methods

### 2.1. Study Area

2.1.1. Spatial Scope of This Study

The study area of this research should be limited so as to improve its efficiency and accuracy. In this study, Nanjing was chosen as the spatial scope for two reasons:

- First, Nanjing is the most typical city to study industrial heritage in China.

Nanjing, known as "the ancient capital of the Six Dynasties", has always been one of the famous cities in China. The establishment of Jinling Arsenal in 1865 marked Nanjing becoming one of the earliest birthplaces of the modern Chinese military industry [72]. On 18 April 1927, Nanjing National Government was established, choosing Nanjing as the capital. The decade from 1927 to 1937, when Nanjing was the capital, was called the "golden decade of modern China". During this period, Nanjing carried out large-

scale urban construction, vigorously promoted industrial development, and laid a good foundation for the urban development of modern Nanjing [73]. By 1936, Nanjing's urban population had increased to over one million, making it one of the six largest cities in China at that time. Therefore, Nanjing has played a very important and wonderful role in the urban development history of modern China, and research on modern Nanjing cities and buildings has always been the focus of many scholars at home and abroad [74–76]. As an important part of modern architecture in Nanjing, modern industrial architecture in Nanjing can not only reflect the developmental characteristics of the modern Chinese industry but also show the distinctive urban culture of modern Nanjing.

- Second, Nanjing has entered a development stage focusing on renewal and redevelopment, and one of its subjects is the large amount of industrial land where modern industrial buildings are located.

With urban expansion and the upgrading and relocation of the manufacturing industry, a large number of industrial enterprises in Nanjing have shut down or moved away. Due to operating costs, pollution control, industrial elimination and other factors, a large number of industrial buildings or land have been abandoned. However, the Nanjing Municipal Government did not approve the Nanjing Industrial Heritage Preservation Plan until March 2017, with a list of 40 industrial heritage protection sites in Nanjing [77]. Unfortunately, the protection contents and measures of most industrial buildings or sites in the list are not clarified by far. Therefore, the systematic study of the preservation and reuse of industrial heritage in Nanjing is urgently needed.

In order to ensure the comparability of the selected data within a certain period of time, the spatial scope of this paper is defined as the urban boundary of Nanjing in the latest version of the map of Jiangsu Province issued by the People's Government of Jiangsu Province in April 2021. Here, the urban planning situations of Nanjing in various periods can be analyzed accurately.

### 2.1.2. Temporal Scale of This Study

The statistical data related to this study published by the Nanjing government are mainly concentrated during and after the 1990s, which is the golden period of development after China's reform and opening up. Therefore, in order to ensure the coverage and accuracy of the data, the time range of this study was set to the period 1990–2020.

### 2.1.3. Definition of Research Object

Industrial heritage is a new concept, which is an extension of ancient cultural heritage. In this regard, this post-industrial activity must not appear as an independent development in a nation's history but a follow up in order to be sustainable. Combined land use for post-industrial cultural reuse is entangled in earlier cultural periods, but it is sometimes present in the same land use projects [78].

It is necessary to explain the definition and scope of "industrial heritage" in this study. Regarding the question of 'what is industrial heritage?', there has been an authorized definition that is widely accepted at the international level, for example, as articulated in The Nizhny Tagil Charter:

> "Industrial heritage consists of the remains of industrial culture which are of historical, technological, social, architectural, or scientific value. These remains consist of buildings and machinery, workshops, mills and factories, mines and sites for processing and refining, warehouses and stores, places where energy is generated, transmitted and used, transport and all its infrastructure, as well as places used for social activities related to industry such as housing, religious worship or education [ . . . ]. The historical period of principal interest extends forward from the beginning of the Industrial Revolution in the second half of the eighteenth century up to and including the present day, while also examining its

earlier preindustrial and protoindustrial roots. In addition, it draws on the study of work and working techniques encompassed by the history of technology." [79].

It is certain that the words written in The Nizhny Tagil Charter are instrumental in specifying the value, type, content and scale of 'industrial heritage', particularly in the Western context. However, in China, the definition of industrial heritage has been generally evolved. Ning Lu argues that the idea of Chinese industrial heritage has developed on the basis of some assumptions that are deeply embedded in the authorized heritage discourse (AHD) and its various adjuncts, for example, a number of international charters and conventions [38]. In the definition, however, the quality of industrial heritage is inevitably influenced by the specific context of China, which is a feature that is, in part, the result of different social and historical perspectives [80].

In 2006, the growing debate over the lack of a common ground for the definition and evaluation of industrial heritage, together with the continued destruction of underused industrial buildings and sites, resulted in the first national symposium, held in Wuxi, to address these issues. Meanwhile, the first official document—the Wuxi Proposal—was promulgated by the conference committees and the State Administration of Cultural Heritage (SACH), concentrating on the definition and justification of value. Therefore, the idea of Chinese 'industrial heritage' was clearly elaborated in this document for the first time:

> "The industrial heritage contains both the tangible and the intangible industrial remains of historical, sociological, architectural, technological or aesthetic value, including factories, workshops, mills, warehouses, shops and other industrial structures; mines, processing and smelting sites, energy production sites, transmission and usage sites, transportation facilities, social activities sites with industrial production, industrial equipment, production technology, data records, enterprise culture [ . . . ]. Since the First Opium War, there have been various industrial remains left as the legacy of all phases of modern industrial construction in China, which constitute the principal part of China's industrial heritage, witness and record the change and development of modern Chinese society." [81].

Compared with the international definition of industrial heritage as noted above, the Wuxi Proposal defined the concept of industrial heritage in a similar way, for example, with the Nizhny Tagil Charter, but at the same time, it distinguished the historical periods of principal interest starting from the First Opium War (1840–1842) rather than the Industrial Revolution. Furthermore, the proposal highlighted the intangible cultural elements of industrial heritage, which is an extension of the concept of industrial heritage.

Against this background, the time range of the research object in this paper is limited to the period 1865–1949, a period of China's "capitalist modernization", within the period of modern China from 1840 to 1949. To facilitate comparative analysis, the research object has been limited to specific building typologies. Industrial buildings equipped with mechanical power under the influence of Western advanced technology are within the research scope of this paper [15]. In addition, industrial legacy is distinguished from industrial heritage. The key distinction between heritage and legacy lies in the identification of "heritage identity". The existing "industrial legacy", as the historical inheritance of the preindustrial society, is huge in number, but it also has the characteristics of wide distribution and uneven quality. Therefore, this article focuses on the industrial architectural heritage of Nanjing and selects the specific research objects with reference to the list of 51 protected industrial heritage sites published by the Nanjing Municipal Planning and Natural Resources Bureau in 2013. Figure 2 shows the locations of Nanjing in China and the 51 industrial heritage sites within the spatial scope of the study.

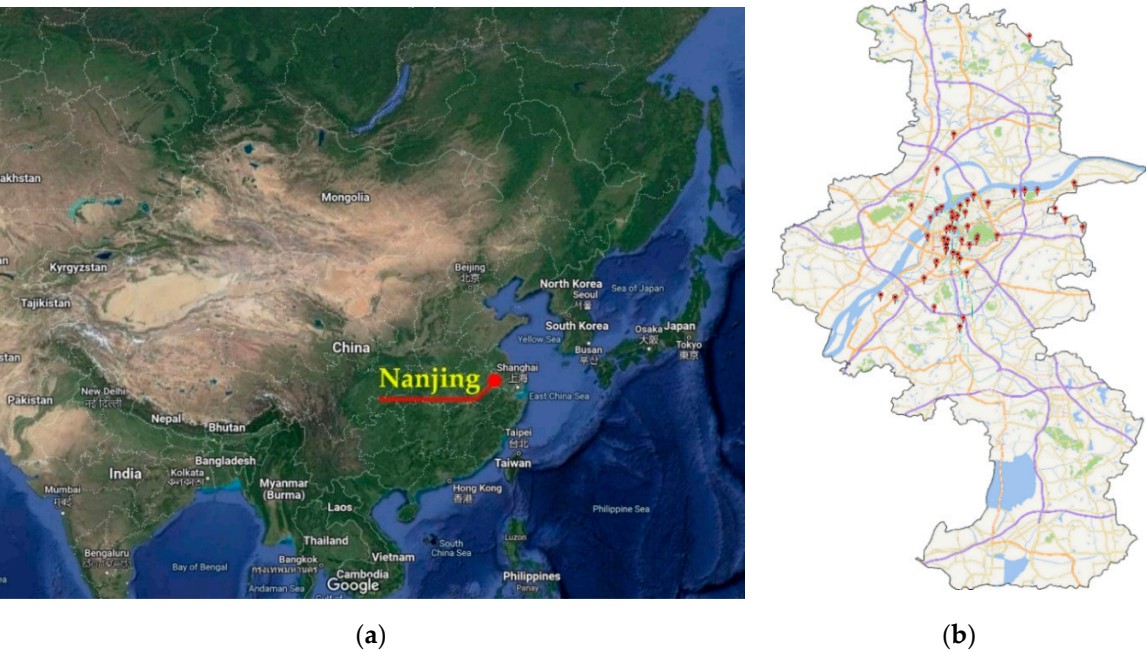

(**a**) (**b**)

**Figure 2.** (**a**) The geographical location of Nanjing in China (Figure source: own drawing; data from: Google Maps); (**b**) Distribution of 51 protected industrial heritage sites in Nanjing (Figure source: own drawing; data from (1) Government department: Nanjing Municipal Planning and Natural Resources Bureau; (2) Site: [82]).

### 2.2. Data

The specific data were collected in four different years (1990, 2000, 2010 and 2020), covering the six factors for protected industrial heritage sites mentioned in Chapter 1. Among them, the data of population density and industrial renovation investment funds were derived from the Nanjing Statistical Yearbook published by Nanjing Municipal Statistics Bureau [83]. The data of industrial heritage sites, transportation and natural resources were derived from the Nanjing Yearbook published by the Nanjing Local Records Compilation Committee [84]. The data of urban growth boundary and eco-environment control area were derived from the Nanjing Municipal Planning and Natural Resources Bureau.

### 2.3. Data Analysis

#### 2.3.1. Statistical Analysis

Nanjing's industrial heritage sites are rich in variety, basically covering the vast majority of industrial types in China. Figure 3 shows that the 51 existing industrial heritage sites in Nanjing represent many industrial sectors.

In order to analyze them more accurately and deeply, the existing industrial heritage sites need to be classified. After consulting the literature, it was concluded that classification by production factors suited our study the best. Production factors mainly refer to social resources required by social production and business activities. Urban planning usually maintains the operation of the national economy through the distribution of various production factors, thus further affecting industrial production activities.

According to the proportion of resources, labor, capital, knowledge and other production factors or the degree of dependence on each factor, the 51 industrial heritage sites were categorized into four categories: labor-intensive, capital-intensive, technology-intensive and resource-intensive. Approximately 40% of the industrial sites in Nanjing can be regarded as capital intensive, while the other categories each account for 20% (Figure 4).

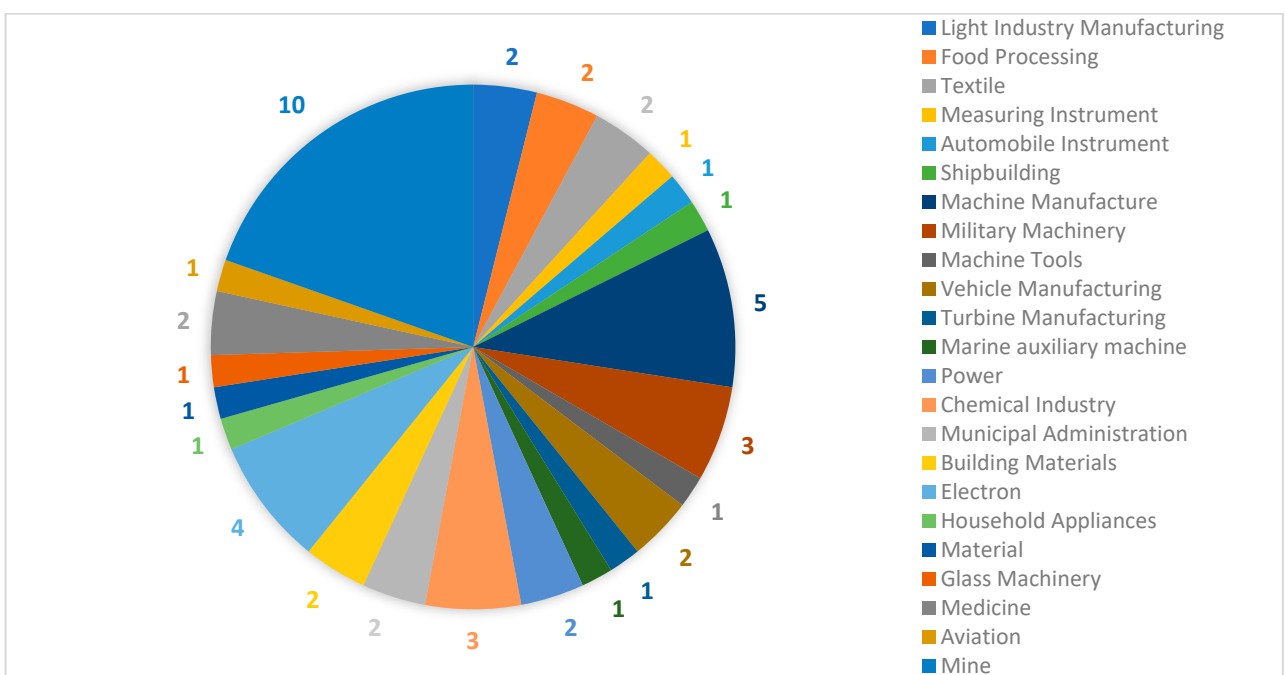

**Figure 3.** Different industrial types amongst Nanjing's 51 industrial heritage sites (Figure source: own drawing. Data from: (1) Government department: Nanjing Municipal Planning and Natural Resources Bureau; (2) book: [85]).

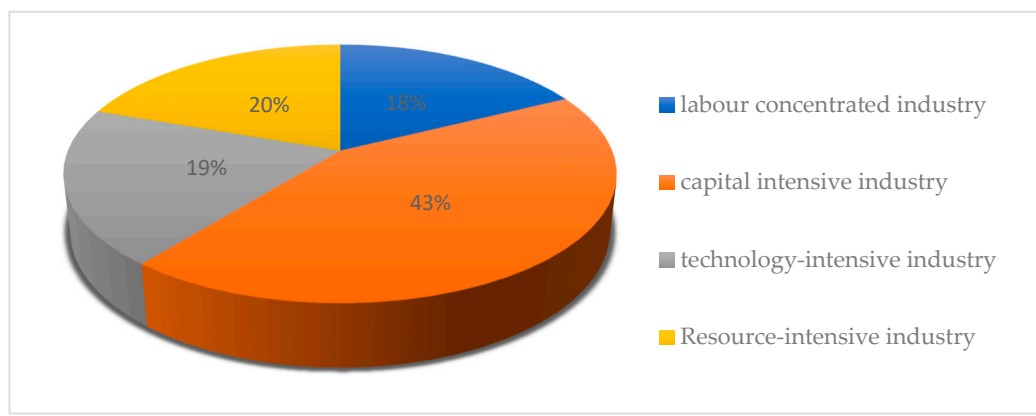

**Figure 4.** Proportion of four industrial categories in Nanjing's 51 industrial heritage sites (Figure source: own drawing).

### 2.3.2. Spatiotemporal Pattern Analysis

Population distribution data for every ten years from 1990 to 2020 were collected, and the population density of each site and period in Nanjing was calculated. For four years (1990, 2000, 2010 and 2020), additional data about the number of sites, original industrial types, development history and current status of 51 industrial heritage sites were collected and analyzed.

### 2.3.3. Identification of Influencing Factors and Explanations of Some Terms

In order to answer the main research question, "how are the processes of industrial site abandonment and regeneration affected by urban planning?", the influencing factors behind it should be identified and discussed first.

- Urban growth boundary

After experiencing the rapid expansion in the early stage, the focus of Nanjing's urban construction gradually shifted from the scale and speed of development to the efficiency and quality of development. Based on the Yangtze River and the historical urban area, the multi-structure, interval distribution and multi-center spatial characteristics were formed (Figure 5). To some extent, the accelerated construction of Nanjing New District inhibited the further agglomeration of the old urban area. However, as the surrounding new districts have not yet formed a perfect comprehensive service function, the historical urban area is still under great pressure. While the environmental and traffic pressure of Downtown Nanjing is still increasing, the economic structure of the historical urban area is transforming from an industry to a service focus. In this process, a large number of manufacturing enterprises in Nanjing have moved to the suburbs, leaving behind many vacant industrial buildings [86]. Therefore, the growth of the urban boundary in Nanjing is an important factor in the processes of industrial heritage abandonment and regeneration.

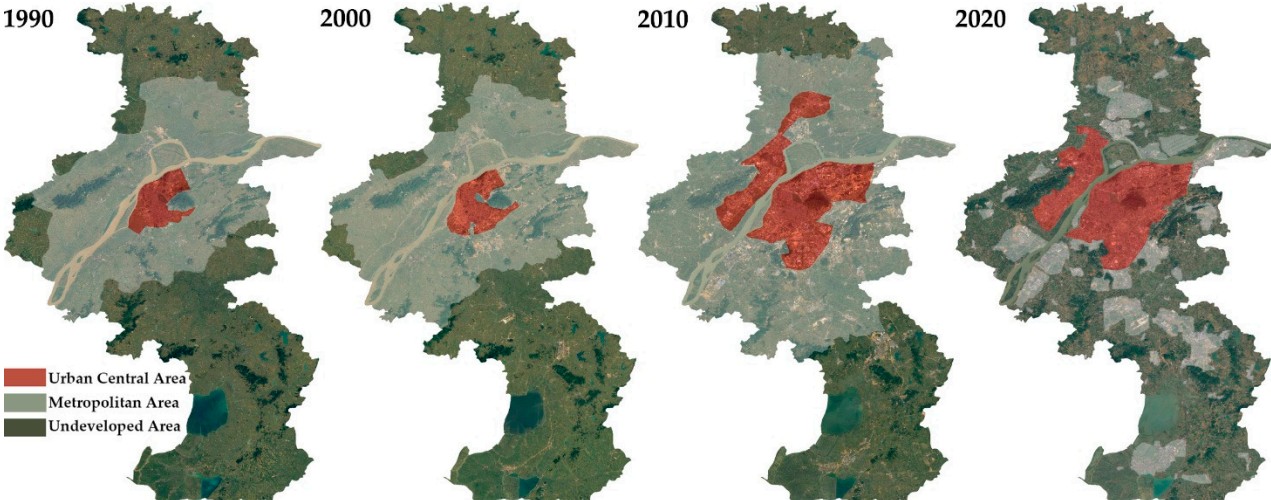

**Figure 5.** The change in boundary of urban growth in Nanjing over time (Figure source: own drawing; data from: [87–89]).

The urban growth boundary (UGB), as one of the policy tools of spatial growth management, can play an important role in coordinating the protection and development needs of a city. It can not only improve the performance of urban internal space through "smart growth" but also protect the urban ecological environment via "ecological priority".

At present, although China and the West have a different understanding of the concept of the urban growth boundary, there are some consensuses as follows [90]:

1.    The urban growth boundary is a multi-objective planning tool to control urban space, aiming to maximize ecological, economic and social benefits, whilst trying to guide urban development in suitable areas; avoid risk areas; and protect ecologically sensitive areas, such as forest land, water area and farmland. At the same time, it can optimize the use of infrastructure and public service facilities by combining the concept of compact growth.

2.    The "boundary" of urban growth is not a fixed boundary. The permanent growth boundary and dynamic growth boundary can be defined according to the needs of urban development. While urban areas are growing rapidly in China, urban growth not only needs to delimit the "rigid" red line of the permanent undeveloped protected area, but it also needs to deal with the elastic "dynamic" boundary for unpredictable development in the suburban district.

The concept of the "boundary" cited in this paper represents an "elastic" growth boundary. The terminology of the UGB mentioned in this paper mainly refers to a tool

of urban planning, which can affect the processes of industrial site abandonment and regeneration.

- Traffic accessibility

In modern and contemporary times, the transportation of the Yangtze River and highways has greatly saved the transportation time and economic cost of industrial materials and thus further affected the location choice and operation state of industrial enterprises in Nanjing [15].

In 1959, W.G. Hansen [91] defined the concept of accessibility for the first time, the interaction opportunity of each node in the traffic network, and established the Hansen potential energy model to calculate the accessibility index. Since then, different scholars have given different definitions of accessibility. After completion of the literature review, we found that accessibility can be divided into two categories according to different definitions: One is the convenience of transportation, and the main measurement standard is time distance or space distance: the shorter the distance, the better the accessibility. The other is the opportunity for interaction [92].

Upon observing the satellite map, we found that the arterial road system in Nanjing was basically completed in 1990, with no obvious change in the following 30 years, while the shape of the Yangtze River in Nanjing has remained unchanged for last 30 years (Figure 6). Thus, we decided to test the convenience of transportation of each industrial site in Nanjing.

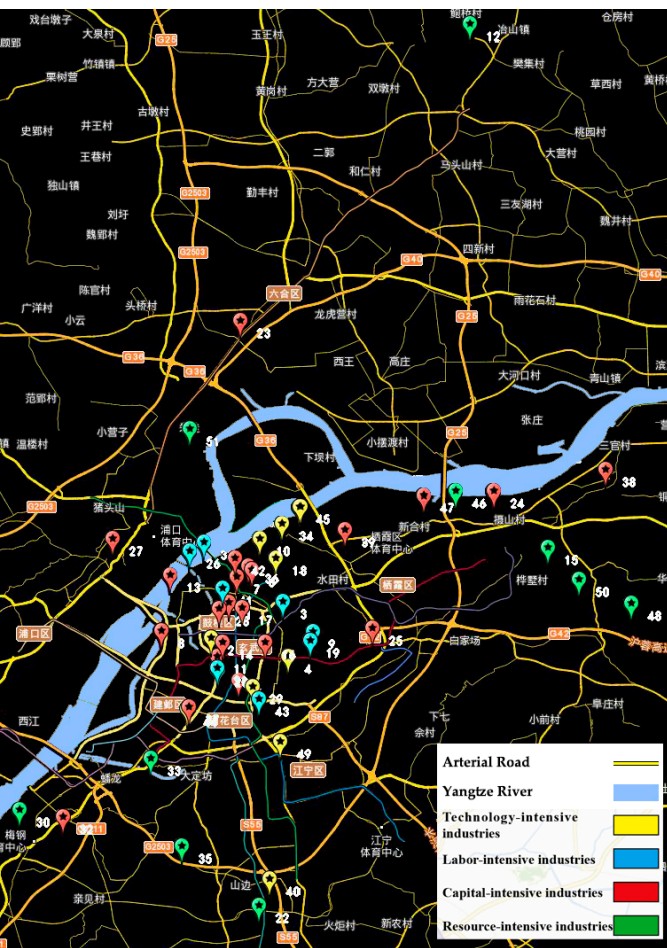

**Figure 6.** The relationship between the four industrial types of Nanjing Industrial Heritage Sites, arterial roads and Yangtze River shoreline (Figure source: own drawing; data from: [82]).

Specifically, this study mainly conducted a comparative analysis of the distances between each industrial heritage site and its nearest arterial road or the shoreline of the

Yangtze River. Among the two distances from each site to its nearest arterial road and to its nearest shoreline of the Yangtze River, we chose the shortest traffic distance as the traffic accessibility index of each heritage site.

- Population distribution

The industrial spatial layout of Nanjing always takes into account the population distribution. In the modern history of Nanjing, some small-scale private enterprises paid more attention to the regional development degree and the acquisition of labor resources when choosing a site. However, some enterprises, which produce noise and dust, choose to build factories in the northwest area of Gulou, where land prices and the number of residents are low [93] (Figure 7). In addition, with the expansion and population growth of Nanjing, many factories that were originally located in the suburbs are gradually approaching new residential areas. In this case, the operation state of the original industrial enterprises is bound to be affected by the change in population distribution.

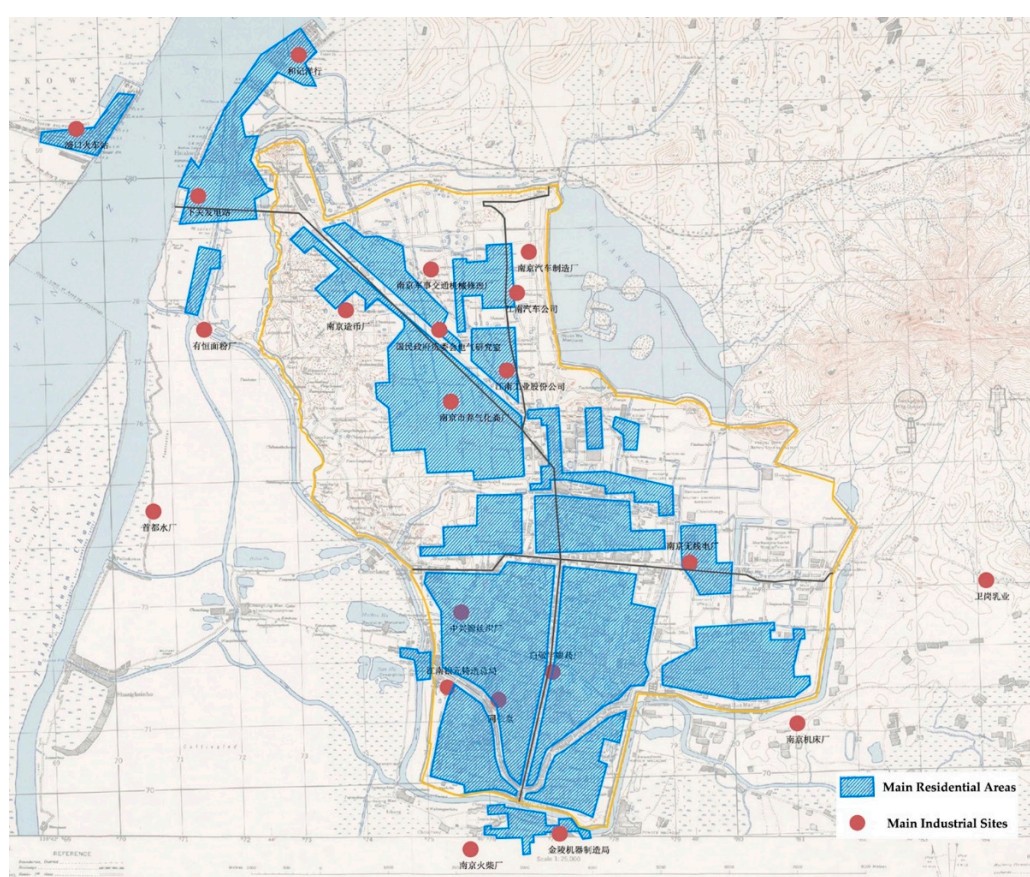

**Figure 7.** Population distribution of Nanjing in 1945 (blue) and main industrial sites (red dot) (Figure from: [15]; data from: Nanjing Archives).

Therefore, specific analysis of the impact of population distribution is also necessary. In this article, we used the population density index, which has been widely used, to analyze the population distribution around each industrial site. The unit of population density in this paper was "persons/km$^2$".

- Industrial renovation investment

Industrial production activities cannot efficiently operate constantly without the financial support of the whole society. Industrial renovation funds are very important for industrial equipment purchase and maintenance, building renovation and construction. Undoubtedly, the investment in industrial renovation has an impact on the abandonment and regeneration processes of Nanjing industrial heritage sites to a certain extent.

In this article, we used "industrial fixed assets renovation fund" as the index of industrial renovation investment. It is also the general name of all funds used for the renovation of fixed assets and the technical performance of existing industrial enterprises.

- Natural resource change

The raw materials or mineral resources needed for industrial processing often determine the industrial site selection. The site selection of some historical industries in Nanjing not only requires a proper distance between mineral resources and factories but also has high requirements for the reserves and quality of mineral resources [94–97]. Therefore, natural resources constitute an important factor affecting the distribution of industrial sites in Nanjing.

In the analysis of "natural resource change", we used the amount of abandonment and reuse caused by environmental deterioration as the index to measure the degree of influence of natural resources.

- Eco-environmental policies

Over the past 30 years, the ecological environment has been damaged by Nanjing's urban construction to a certain extent, especially by industrial enterprises. The eco-environmental red line has been delimited many times by the Nanjing government in order to preserve the eco-environment as much as possible (Figure 8). On the other hand, the increasing concern regarding serious environmental problems has also led to the shutdown and relocation of a large number of industrial sites in the last 30 years.

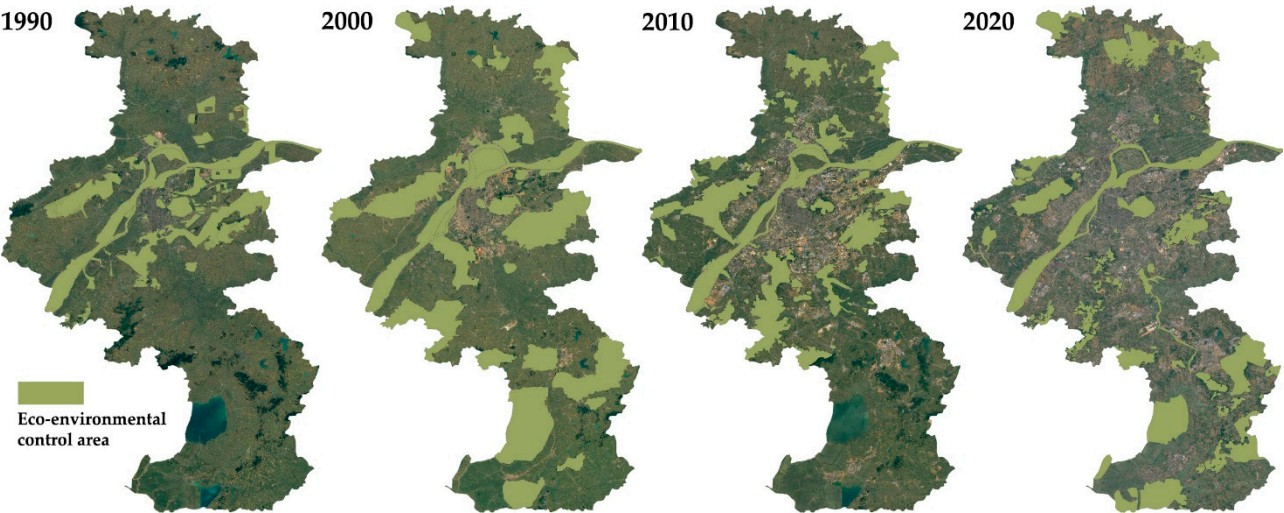

**Figure 8.** The change in eco-environmental control area in Nanjing over time (Figure source: own drawing; data from: [87–89]), (As the early urban expansion had not yet spread to the suburbs, the urban masterplans in 1990 and 2010 only planned for ecological conservation areas in metropolitan areas but did not clearly mark them in suburbs).

In this study, we used the ecological protection red line to show the eco-environmental policies of Nanjing. The essence of the ecological protection red line is the line of ecological environmental security. The aim of this is to guarantee ecological function, environmental quality and effective natural resource utilization, thus further promoting the balance of the population, resources and the environment and the unification of economic, social and ecological benefits. In the analysis, the relationship between the proportion of abandonment and reuse of industrial sites and the ecological protection red line was analyzed, which intuitively revealed the influence of ecological policy.

According to the analysis and discussion above, we propose the following hypothesis: on the urban scale, the processes of industrial heritage site abandonment and regeneration in Nanjing since the 1990s may be affected by six influencing factors, namely, the urban

growth boundary, traffic accessibility, eco-environmental policies, population distribution, industrial renovation investment and natural resource change. Figure 9 represents the possible framework of interpretation of this influencing mechanism.

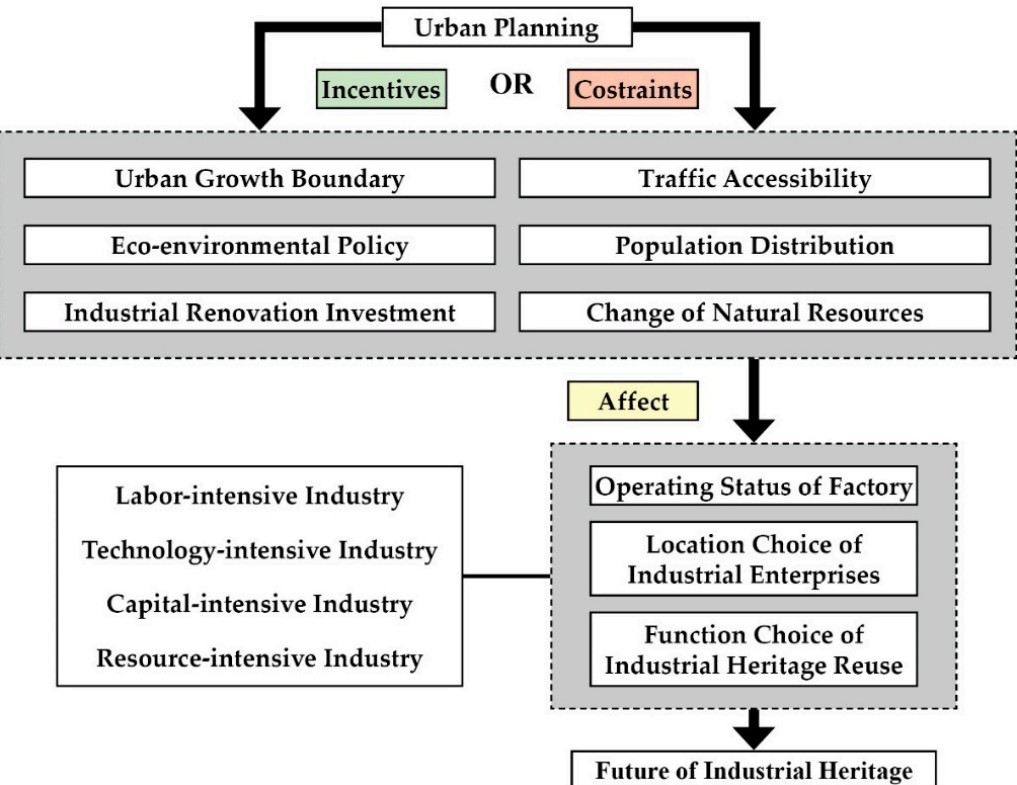

**Figure 9.** Framework of interpretation of the influencing mechanism (Figure source: own drawing).

Additionally, in order to avoid repeated explanations and prolix expressions, the article also used some terms to summarize some phenomena and development processes:

- Operation state

In this article, this refers to the state of industrial heritage at a certain moment, including three states: the original industry still in operation, abandoned or reused for other purposes.

- Function choice of industrial heritage reuse

In this article, this means the reuse purpose of original industrial buildings or sites.

- Urban central area

"Urban central area" in this article refers to the main urban area, and it also includes adjacent functional groups and areas that need strengthened land use control.

- Metropolitan area

The metropolitan area is the most commonly used regional concept of urban function in Western countries. It is a combination of a large population core area and neighboring communities, which are socially and economically integrated.

### 3. Results

#### 3.1. Influence of Different Factors on the State of Labor-Intensive Industrial Heritage Sites

The labor-intensive industry refers to an industry that mainly relies on a large amount of labor for production but has low dependence on technology and equipment. The measurement standard is whether wages and equipment depreciation account for a larger proportion of the production cost than research expenditure.

From Figure 10, we can see that the operation state of labor-intensive industrial heritage sites in Nanjing did not change significantly in the early stage of Nanjing's rapid urban expansion. However, in the last 20 years, six-ninths of the original factories of labor-intensive industrial heritage sites were abandoned or reused as service industries. This shows that the UGB has a considerable impact on the operation state of labor-intensive industries.

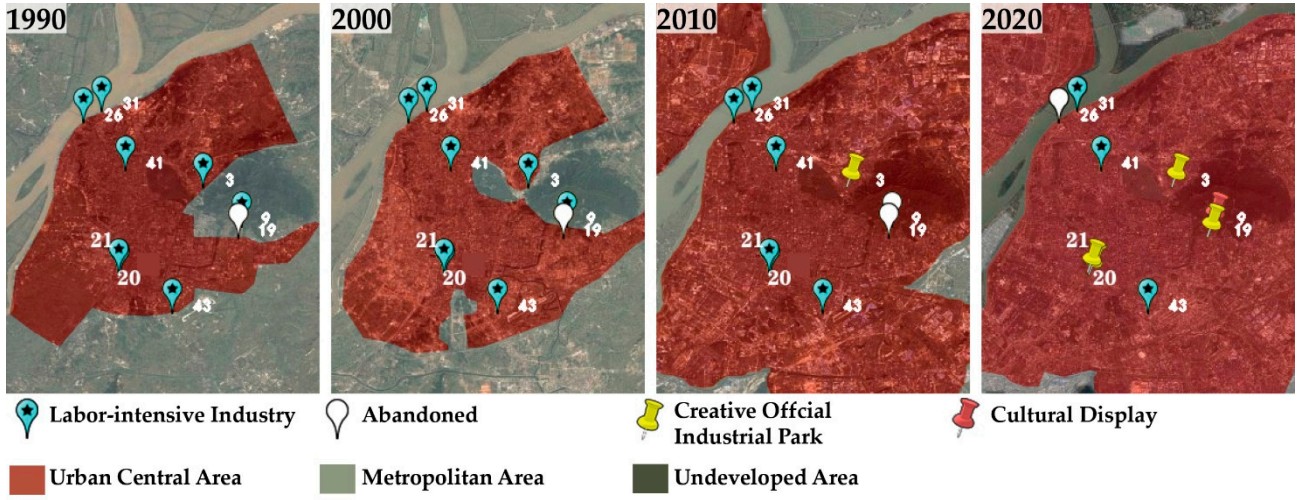

**Figure 10.** The change in the relationship between the operation state of labor-intensive industries and urban growth boundary over time (Figure source: own drawing; data source: on-site investigation and interview).

The accessibility analysis of labor-intensive industrial heritage sites can be found in Table 1. Statistics show that the traffic accessibility of labor-intensive industrial heritage sites is positively correlated with "whether it is abandoned or reused", which indicates that unobstructed traffic directly accelerates the processes of abandonment and regeneration of labor-intensive industries.

**Table 1.** Statistics of accessibility analysis of labor-intensive industrial heritage.

| Site Number | Traffic Accessibility [1] | Is It Abandoned or Reused? |
| --- | --- | --- |
| No. 43 | 1.15 km | No |
| No. 20 | 0.64 km | Yes |
| No. 3 | 0.56 km | Yes |
| No. 21 | 0.56 km | Yes |
| No. 9 | 0.38 km | Yes |
| No. 19 | 0.17 km | Yes |
| No. 41 | 0.08 km | Yes (change from manufacturing to industrial equipment sales) |
| No. 26 | 0 km | Yes |
| No. 31 | 0 km | Yes (plan to relocate in 2023) |

[1] Accessibility data are the shortest traffic distance from each site to its nearest aerial road or Yangtze River shoreline.

As shown in Figure 11, five labor-intensive industrial heritage sites were always in or close to the eco-environmental control area. Among them, the No. 19 industrial heritage site has long been abandoned. Three industrial heritage sites, No. 3, No. 9, and No. 26, were relocated or repurposed as service industries during the period 2000 to 2020, while the No. 31 industrial heritage site was required by the government to start relocation in 2023 (quoted from recent news in Nanjing). This indicates that Nanjing's ecological protection policy directly affects the operation state of labor-intensive industrial heritage sites.

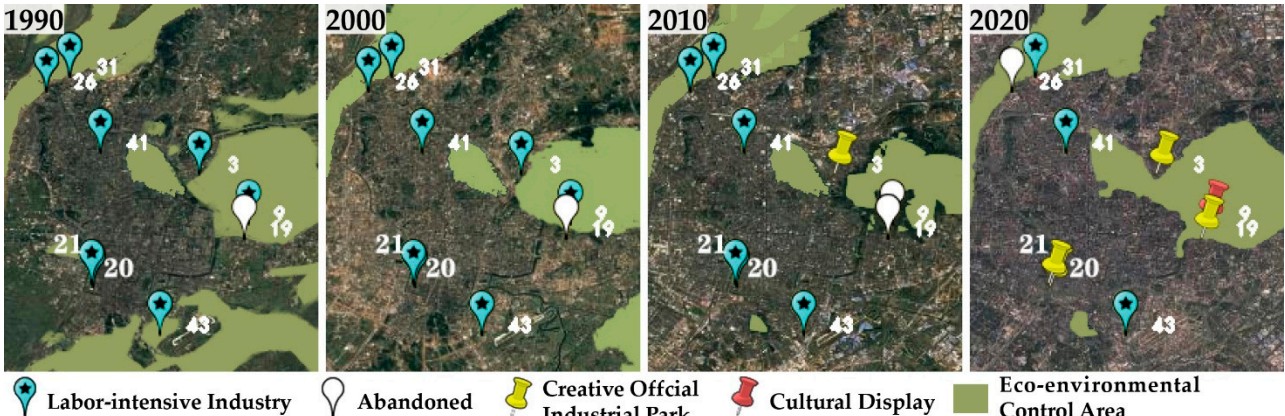

**Figure 11.** The change in the relationship between the operation state of labor-intensive industries and the eco-environmental control area over time (Figure source: own drawing; data source: on-site investigation and interview).

By accounting for the population density around Nanjing's existing labor-intensive industrial heritage sites, we calculated the average population density index of this group over four years. Figure 12 shows that while the population density fluctuated greatly in the period from 1990 to 2010, the operation state of labor-intensive industrial heritage sites changed significantly only between 2010 and 2020. This indicates that the operation state of Nanjing's labor-intensive industrial heritage sites may be mainly influenced by other factors.

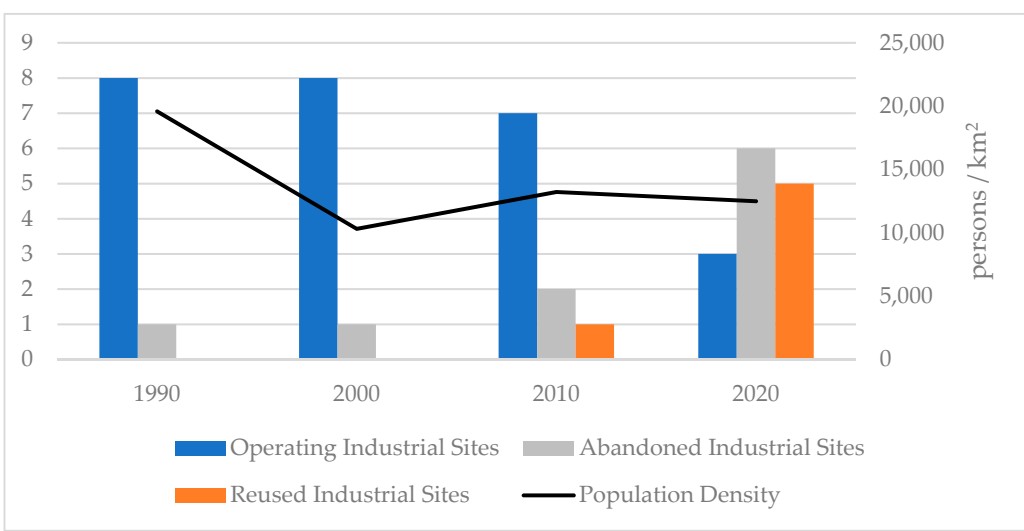

**Figure 12.** The change in the relationship between the operation state of labor-intensive industries and the population density over time (Figure source: own drawing; data from: [84]).

In Figure 13, neither the amount of industrial renovation investment funds nor the operating state of labor-intensive industrial heritage sites showed an obvious change from 1990 to 2000. However, in the period from 2000 to 2010, the amount of investment funds for industrial renovation increased rapidly, while the operation state of labor-intensive industrial heritage sites changed slightly. In the past ten years, the amount of investment funds has dropped significantly, while six-ninths of the labor-intensive industrial heritage sites have been shut down, and five-ninths of the industrial sites have been reused. This indicates that the investment in industrial renovation has a considerable impact on the operation state of labor-intensive industrial heritage sites.

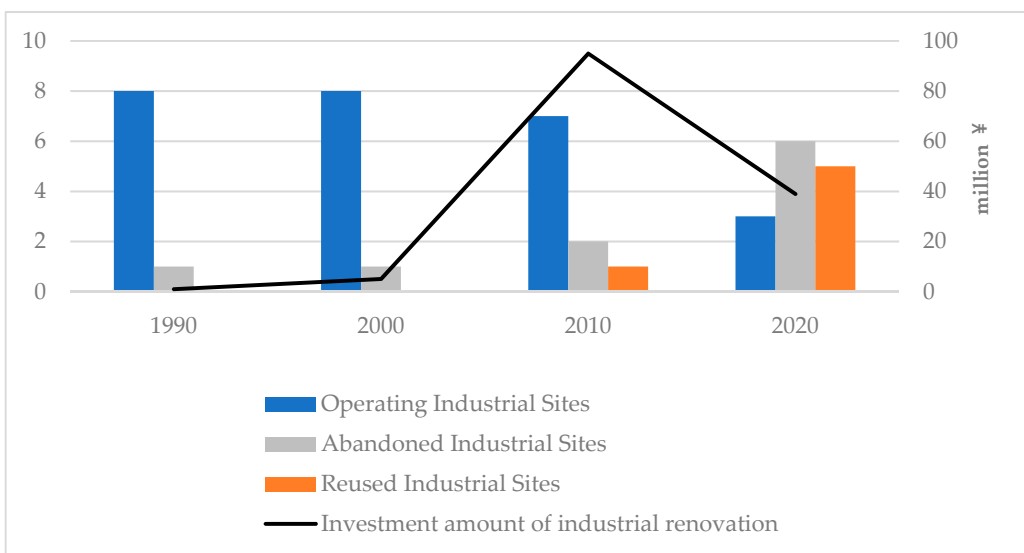

**Figure 13.** The change in the relationship between the operation state of labor-intensive industries and the investment amount of industrial renovation over time (Figure source: own drawing; data from: [83]).

By on-site interviews, we counted the cases of industrial heritage site abandonment or reuse caused by the change in natural resources. Pie Figure 14 shows that none of the abandonment or transformation cases was caused by the deterioration of natural resources. Therefore, the change in natural resources is not related to their operation state.

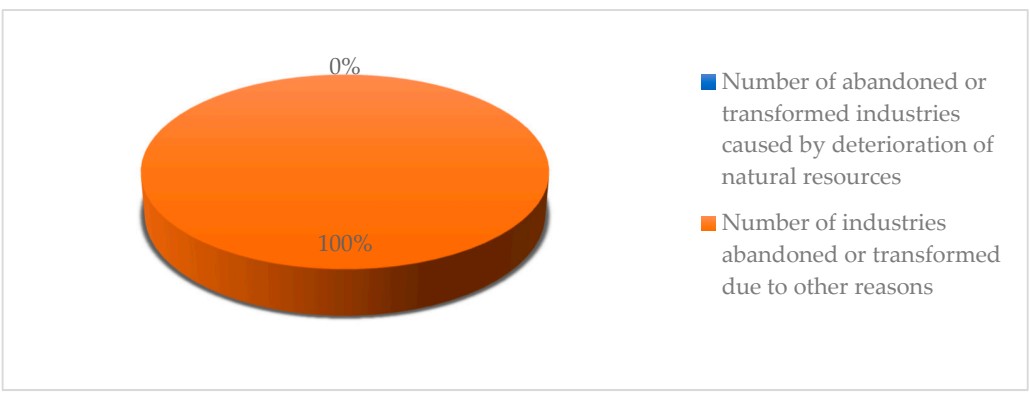

**Figure 14.** The relationship between the operation state of labor-intensive industries and natural resources deterioration (Figure source: own drawing; data source: on-site interview).

*3.2. Influence of Different Factors on the State of Capital-Intensive Industrial Heritage Sites*

Capital-intensive industries refer to industries that need more capital investment. A common feature of them is abundant technical equipment, massive investment, less labor, slow capital turnover and slow investment effect. Capital-intensive industries are mainly distributed in basic and heavy processing industries, and they are usually regarded as an important foundation for developing the national economy and realizing industrialization.

From Figure 15, we can see that during the period from 1990 to 2010, almost all the abandoned or reused capital-intensive industrial heritage sites were in the city center, which reflected the social background that the economic structure in this historical area adjusted to a large extent during this period. In the past ten years, an increasing number of capital-intensive industries have been forced to shut down or migrate with the expansion of the central urban area and the formation of satellite towns. Thus, the operation state of capital-intensive industrial heritage sites is directly affected by the UGB.

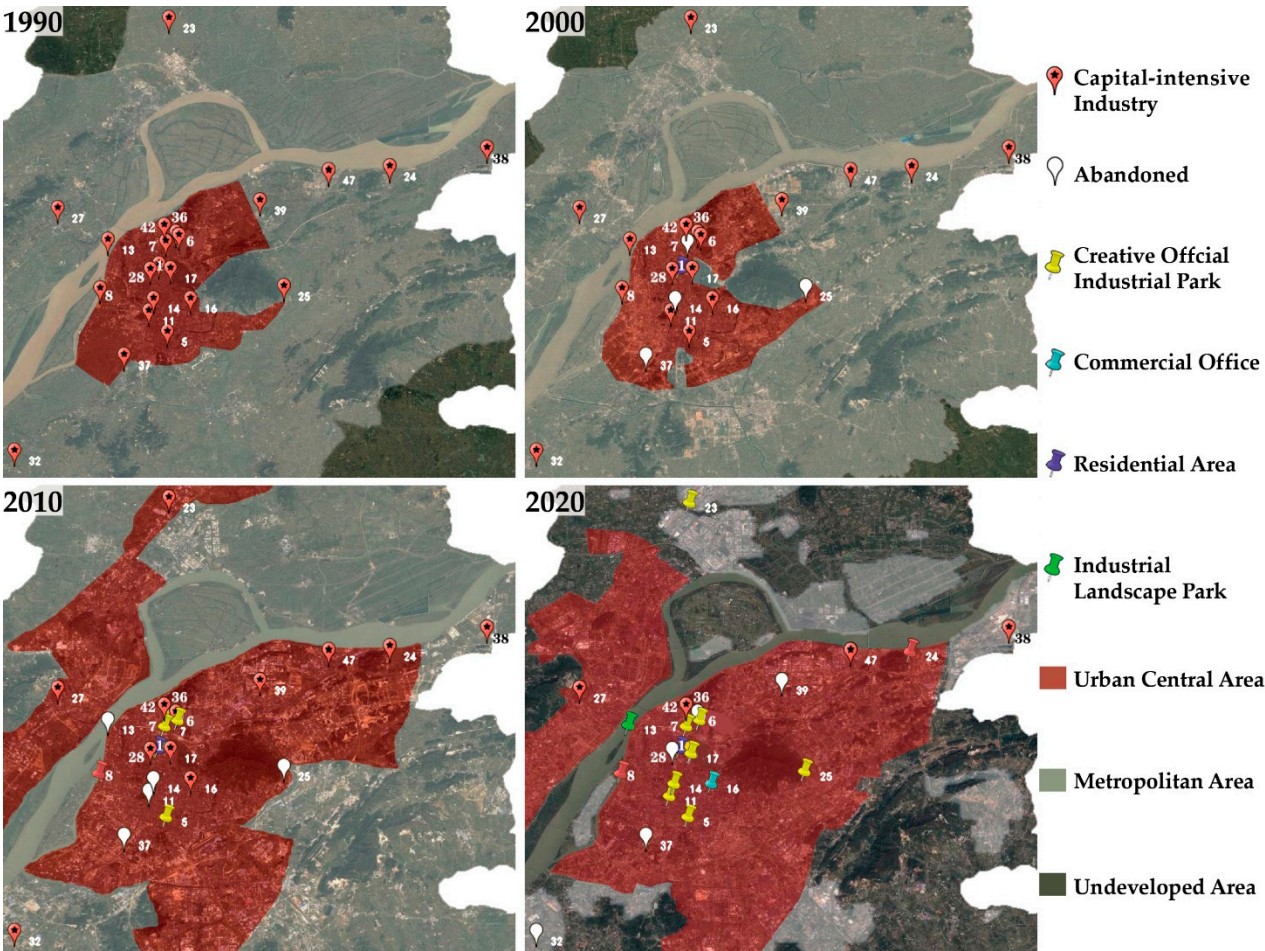

**Figure 15.** The change in the relationship between the operation state of capital-intensive industries and urban growth boundary over time (Figure source: own drawing; data source: on-site investigation and interview).

The accessibility analysis of capital-intensive industrial heritage sites can be found in Table 2. Calculated from the statistics, the average shortest traffic distance of abandoned or reused industrial heritage sites is 0.25 km, while that of those still in operation is 0.65 km, which indicates that the traffic accessibility of most capital-intensive industrial heritage sites is good. This indicates that unobstructed traffic has no obvious impact on the process of capital-intensive industry abandonment or regeneration.

Figure 16 displays that, for a long time, there were eight capital-intensive industrial heritage sites in or close to the eco-environmental control area. Among them, six industrial heritage sites, No. 7, No. 8, No. 13, No. 17, No. 24 and No. 25, have been relocated or reused. No. 38 (close to the eco-environmental control area of the neighboring city; cannot be shown in the figure) was forced to shut down by the Nanjing government in 2021 (quoted from recent news in Nanjing). However, No. 27, which is an important pillar of Nanjing's economic development, continues to operate. This indicates that the ecological conservation policy has a great influence on the operation state of capital-intensive heritage.

**Table 2.** Statistics of accessibility analysis of capital-intensive industrial heritage.

| Site Number | Traffic Accessibility [1] | Is It Abandoned or Reused? |
|---|---|---|
| No. 47 | 0.99 km | No |
| No. 42 | 0.67 km | No |
| No. 27 | 0.28 km | No |
| No. 24 | 0.8 km | Yes |
| No. 37 | 0.66 km | Yes |
| No. 5 | 0.61 km | Yes |
| No. 17 | 0.59 km | Yes |
| No. 16 | 0.33 km | Yes |
| No. 38 | 0.24 km | Yes (forcibly closed by the government in 2021) |
| No. 1 | 0.22 km | Yes |
| No. 25 | 0.22 km | Yes |
| No. 23 | 0.22 km | Yes |
| No. 32 | 0.2 km | Yes |
| No. 8 | 0.12 km | Yes |
| No. 7 | 0.11 km | Yes |
| No. 36 | 0.1 km | Yes |
| No. 6 | 0.06 km | Yes |
| No. 11 | 0.06 km | Yes |
| No. 39 | 0.06 km | Yes |
| No. 14 | 0.05 km | Yes |
| No. 28 | 0.01 km | Yes |
| No. 13 | 0 km | Yes |

[1] Accessibility data are the shortest traffic distance from each site to its nearest aerial road or Yangtze River shoreline.

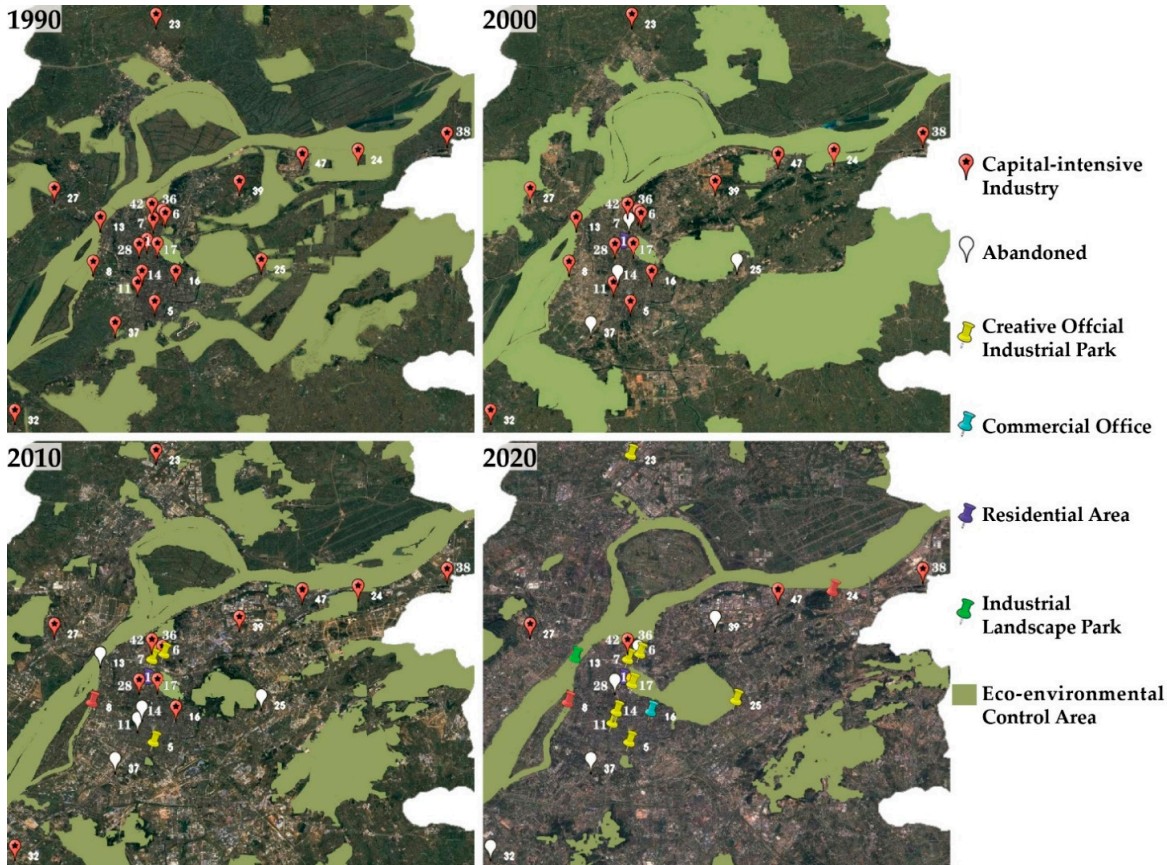

**Figure 16.** The change in the relationship between the operation state of capital-intensive industries and the eco-environmental control area over time (Figure source: own drawing; data source: on-site investigation and interview).

What is worth mentioning is that, nearly half of the existing capital-intensive industrial heritage in Nanjing is located in the old town. Capital-intensive industries usually have huge volume and pollution problems, which may have a negative impact on the cultural heritage relics' preservation in the old town. Figure 17 shows that all the capital-intensive industrial heritages located in the old town have been abandoned or reused during 30 years, which indicates that cultural heritage relics have a great influence on capital-intensive industries' operation.

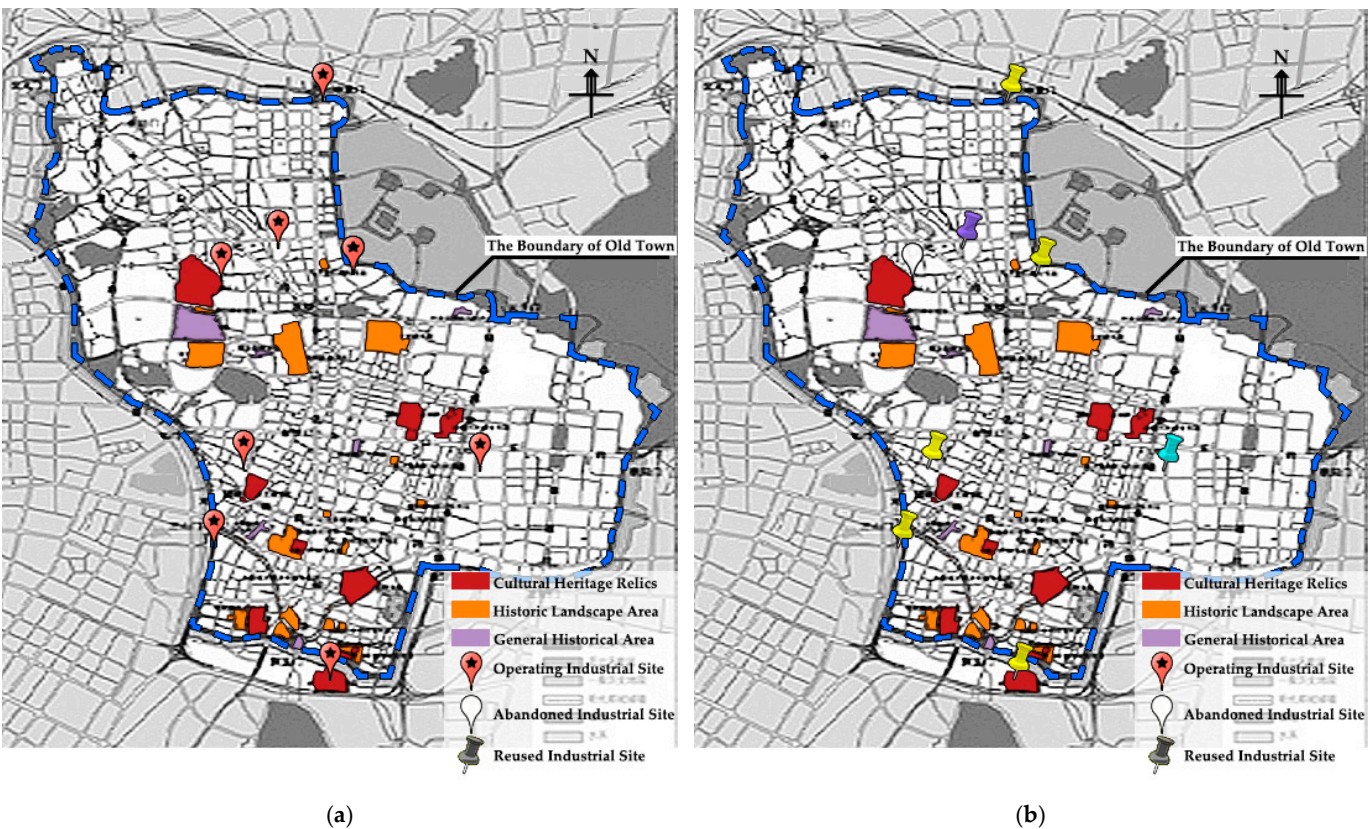

(**a**)  (**b**)

**Figure 17.** The relationship between the operation state of capital-intensive industries and the cultural heritage relics changing with time: (**a**) 1990; (**b**) 2020, (Figure source: own drawing, data from: [98]).

By accounting for the population density around Nanjing's existing capital-intensive industrial heritage sites, the average population density of this group over four years was calculated. In Figure 18, during the period from 1990 to 2000, either the operating number of capital-intensive industrial heritage sites or the population density around it decreased significantly. In the last 20 years, more migrants have poured into Nanjing, which has led to the gradual rise of original capital-intensive industrial heritage site abandonment and reuse. This shows that population distribution has a considerable impact on the operation state of capital-intensive industrial heritage sites.

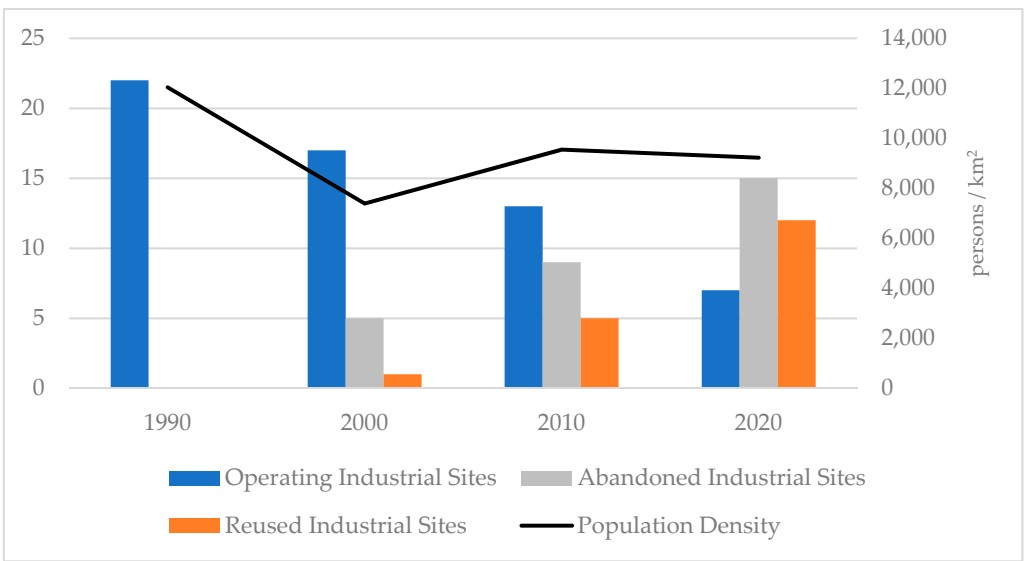

**Figure 18.** The change in the relationship between the operation state of capital-intensive industries and the population density over time (Figure source: own drawing; data from: [84]).

Figure 19 demonstrates that the operating quantity of capital-intensive industrial heritage sites has gradually decreased in the past 30 years, while the number of abandoned or reused has gradually increased. However, the amount of investment funds for industrial renovation in the whole city has fluctuated greatly in the last 20 years, which shows that the operating state of capital-intensive industrial heritage sites is not directly affected by the amount of investment funds for industrial renovation.

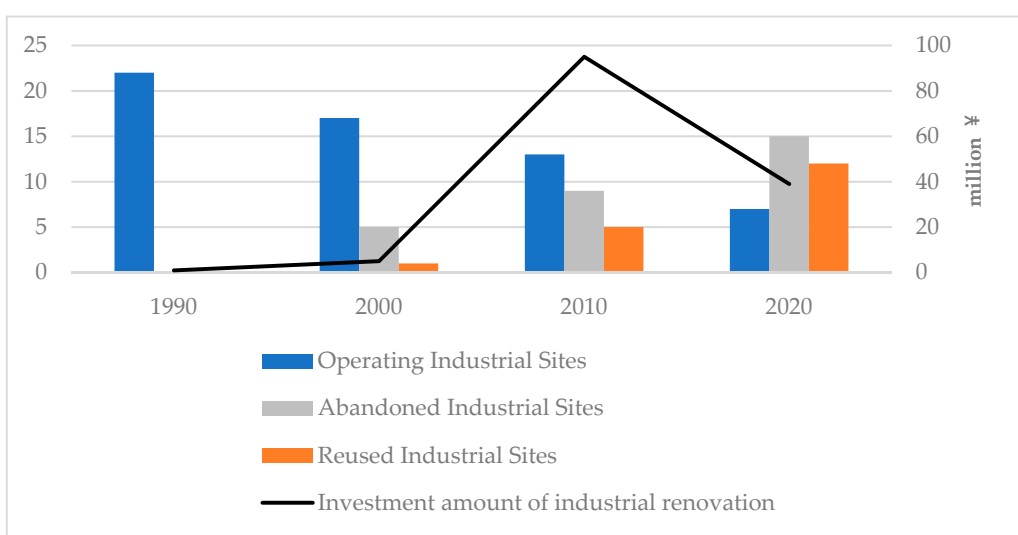

**Figure 19.** The change in the relationship between the operation state of capital-intensive industries and the investment amount of industrial renovation over time (Figure source: own drawing; data from: [83]).

By on-site interviews, the cases of capital-intensive industrial heritage site abandonment and reuse caused by the change in natural resources were counted. The study found that 18% of the cases of abandonment or transformation were related to the deterioration of natural resources (Figure 20). This indicates that the change in natural resources has little impact on their operation state.

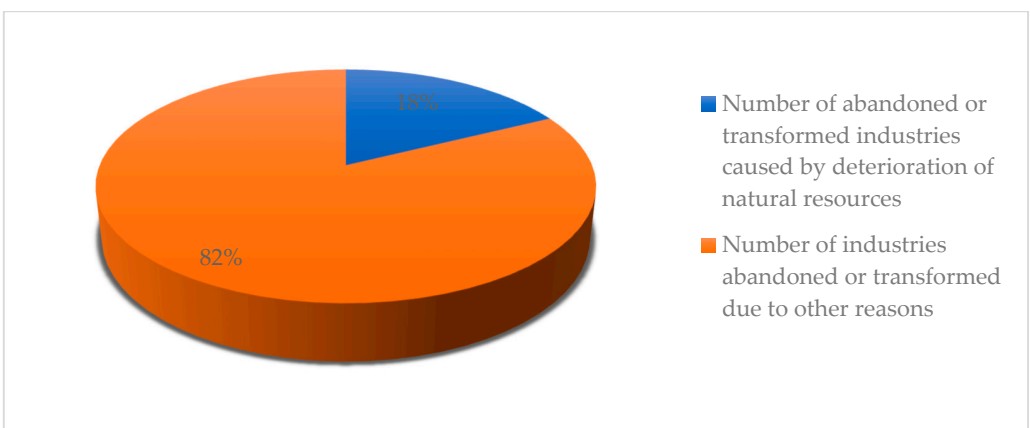

**Figure 20.** The relationship between the operation state of capital-intensive industries and natural resources deterioration (Figure source: own drawing; data source: on-site interview).

*3.3. Influence of Different Factors on the State of Technology-Intensive Industrial Heritage Sites*

Technology-intensive industries, also known as "intelligence-intensive industries", take knowledge capital as the main production factor, mainly providing products and services with intelligence, knowledge, technology, experience, information and skills. Features of their production structure are large proportion technical knowledge, high scientific research costs, workers' cultural and technical level, and high and fast-growing product added value.

From Figure 21, we can see that in the past 30 years, six-tenths of Nanjing's technology-intensive industrial heritage sites have been abandoned or reused, among which five sites are located in the urban central area, which indicates that the UGB has a considerable impact on the operation state of technology-intensive industrial heritage sites.

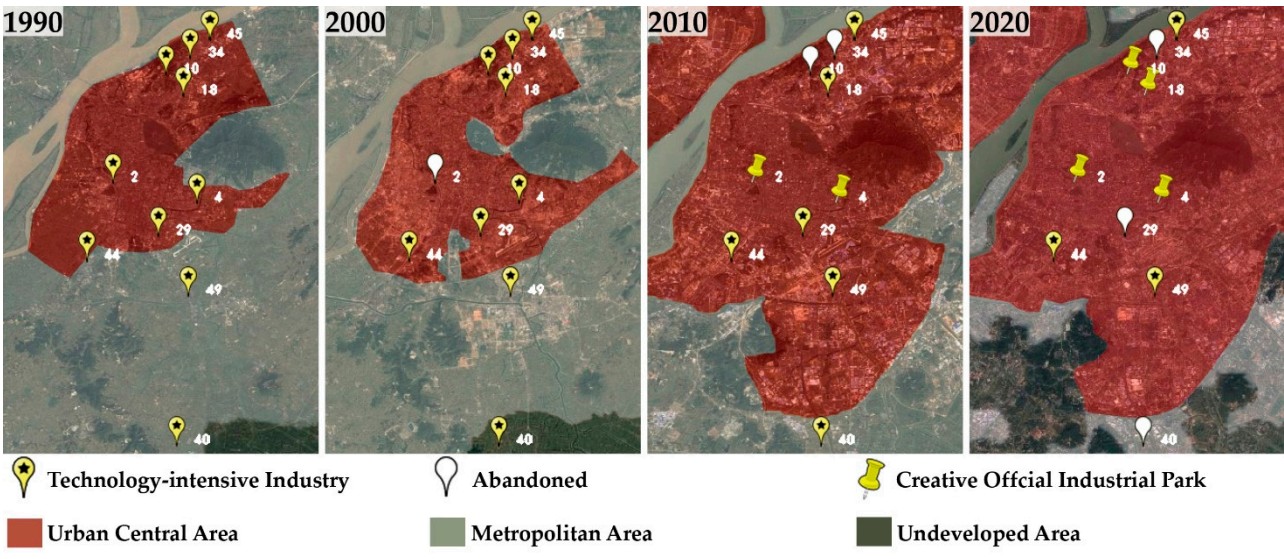

**Figure 21.** The change in the relationship between the operation state of technology-intensive industries and urban growth boundary over time (Figure source: own drawing; data source: on-site investigation and interview).

The accessibility analysis of technology-intensive industrial heritage sites can be found in Table 3. Calculated from the statistics, the average shortest traffic distance of abandoned or reused industrial heritage sites is 0.46 km, while that of those still in operation is 0.09 km, which indicates that the traffic accessibility of most technology-intensive industrial

heritage sites is good. Thus, unobstructed traffic has no obvious impact on the processes of abandonment and regeneration of technology-intensive industries in Nanjing.

**Table 3.** Statistics of accessibility analysis of technology-intensive industrial heritage.

| Site Number | Traffic Accessibility [1] | Is It Abandoned or Reused? |
|---|---|---|
| No. 44 | 0.12 km | No |
| No. 45 | 0.1 km | No |
| No. 49 | 0.06 km | No |
| No. 40 | 1.2 km | Yes |
| No. 4 | 0.88 km | Yes |
| No. 18 | 0.51 km | Yes |
| No. 10 | 0.24 km | Yes |
| No. 2 | 0.18 km | Yes |
| No. 34 | 0.15 km | Yes |
| No. 29 | 0.05 km | Yes |

[1] Accessibility data are the shortest traffic distance from each site to its nearest aerial road or Yangtze River shoreline.

Figure 22 shows that the vast majority of technology-intensive industrial heritage sites in Nanjing are not located within the eco-environmental control area. Therefore, their operation states are not affected by ecological protection policies.

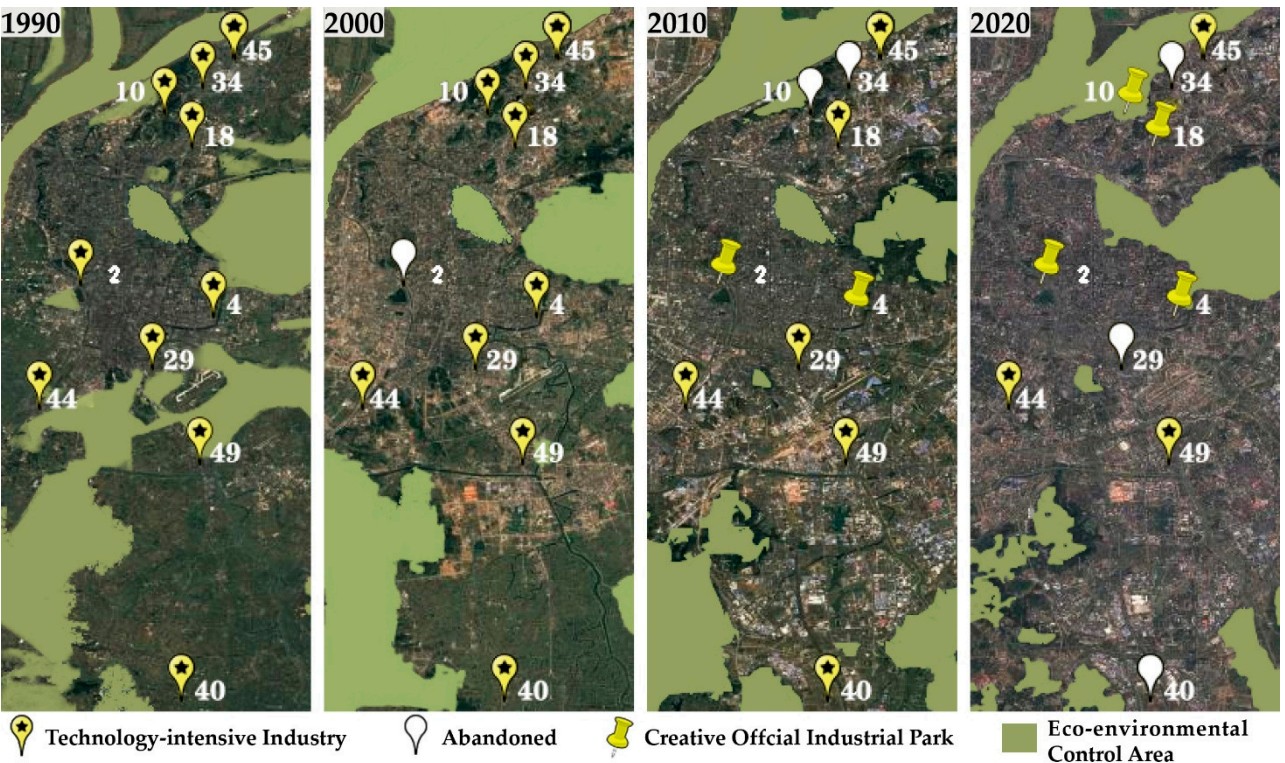

**Figure 22.** The change in the relationship between the operation state of technology-intensive industries and the eco-environmental control area over time (Figure source: own drawing; data source: on-site investigation and interview).

By accounting for the population density around Nanjing's existing technology-intensive industrial heritage sites, the average population density index of this group over four years was calculated. Figure 23 shows that in the past 30 years, the population density around these industrial heritage sites has decreased rapidly, while the operation state of the technology-intensive industrial heritage sites has not changed significantly. In the past 20 years, the population density around these industrial sites and the number of

abandonment and reused sites have shown a slow upward trend. Therefore, the population density has little influence on the operation of technology-intensive industries.

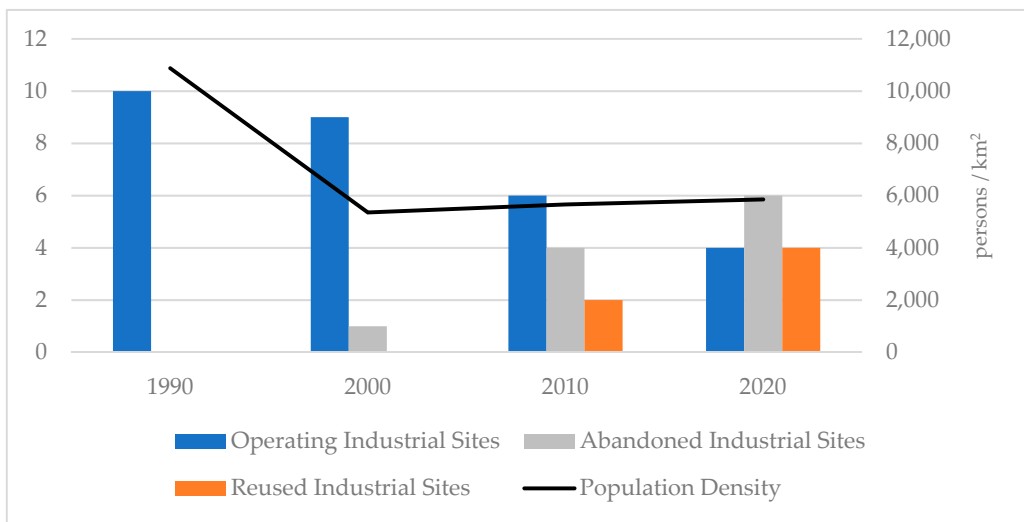

**Figure 23.** The change in the relationship between the operation state of technology-intensive industries and the population density over time (Figure source: own drawing; data from: [84]).

Figure 24 demonstrates that in the past 30 years, the operating quantity of technology-intensive industrial heritage sites has gradually decreased, while the amount that have been abandoned or reused has gradually increased. However, the amount of investment funds for industrial renovation in the whole city has fluctuated greatly in the last 20 years, which indicates that the operation state of technology-intensive industrial heritage sites is not directly affected by the amount of investment funds for industrial renovation.

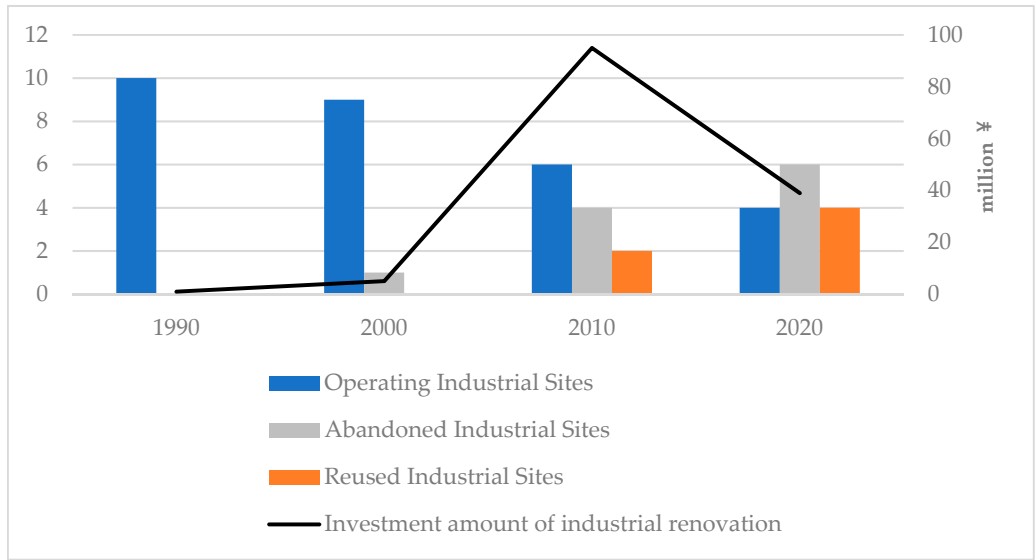

**Figure 24.** The change in the relationship between the operation state of technology-intensive industries and the investment amount of industrial renovation over time (Figure source: own drawing; data from: [83]).

By on-site interviews, the cases of abandonment and reuse of technology-intensive industrial heritage caused by the change in natural resources were counted. Figure 25 shows that 17% of abandoned or reused industrial sites are related to the deterioration of natural resources, which indicates that the change in natural resources has little impact on the operation state of technology-intensive industrial heritage sites in Nanjing.

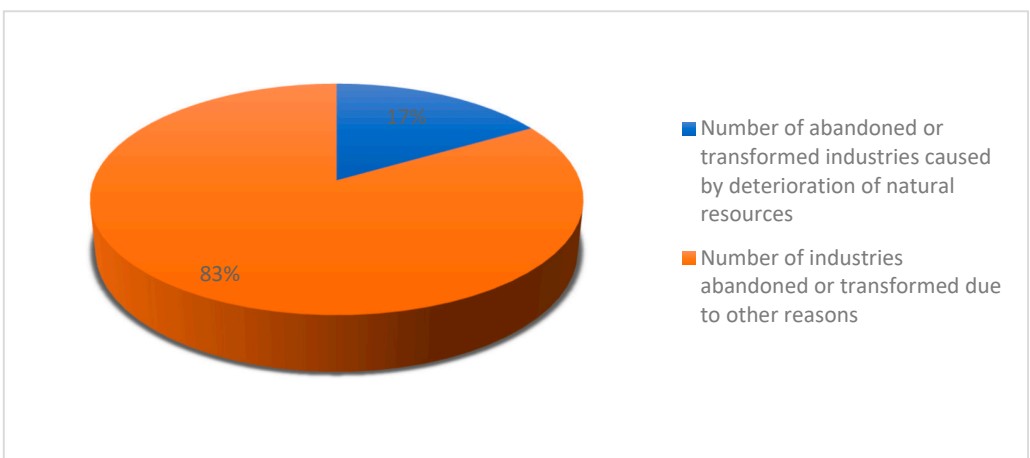

**Figure 25.** The relationship between the operation state of technology-intensive industries and natural resources deterioration (Figure source: own drawing; data source: on-site interview).

### 3.4. Influence of Different Factors on the State of Resource-Intensive Industrial Heritage Sites

Resource-intensive industries are industries that need more natural resources, such as land, for production. As a production factor, "land" here means all kinds of natural resources.

Figure 26 shows that most of the original resource-intensive industries in Nanjing are distributed in urban fringe areas. Therefore, the operation state of resource-intensive industrial heritage sites in Nanjing is not closely related to the change in the UGB.

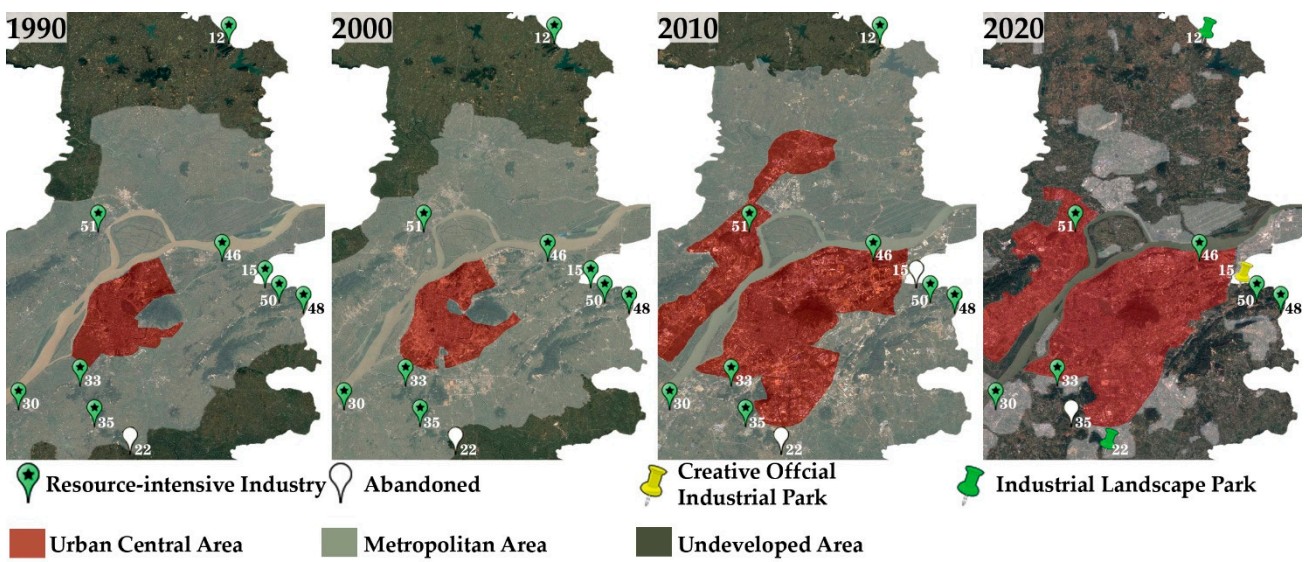

**Figure 26.** The change in the relationship between the operation state of resource-intensive industries and urban growth boundary over time (Figure source: own drawing; data source: on-site investigation and interview).

The accessibility analysis of resource-intensive industrial heritage sites can be found in Table 4. Calculated from the statistics, the average shortest traffic distance of abandoned or reused industrial heritage sites is 2.89 km, while that of those still in operation is 1.21 km. The traffic accessibility of resource-intensive industrial heritage sites is positively correlated with "whether abandoned or reused", which indicates that good traffic accessibility can keep resource-intensive industrial heritage sites from being abandoned or reused to some extent.

**Table 4.** Statistics of accessibility analysis of resource-intensive industrial heritage.

| Site Number | Traffic Accessibility [1] | Is It Abandoned or Reused? |
|:---:|:---:|:---:|
| No. 48 | 2.31 km | No |
| No. 51 | 2.02 km | No |
| No. 33 | 1.58 km | No |
| No. 46 | 0.72 km | No |
| No. 50 | 0.47 km | No |
| No. 30 | 0.16 km | No |
| No. 15 | 6.58 km | Yes |
| No. 22 | 3.06 km | Yes |
| No. 12 | 1.85 km | Yes |
| No. 35 | 0.08 km | Yes |

[1] Accessibility data are the shortest traffic distance from each site to its nearest aerial road or Yangtze River shoreline.

Figure 27 shows that for a long time, almost all resource-intensive industrial heritage sites were in or close to the eco-environmental control area. The 30-year change in the map shows that the eco-environmental control area where these resource-intensive industries are located has obviously narrowed, which indicates that they have caused great damage to the ecological environment. Among them, No. 12, No. 15, No. 22 and No. 35, four resource-intensive industrial heritage sites, have been abandoned or reused as parks or art creation studios, while No. 30 and No. 33 will require closure or transformation within the period from 2020 to 2030 (quoted from recent news in Nanjing). Thus, the ecological environment also has a great influence on the operation state of resource-intensive industrial heritage sites.

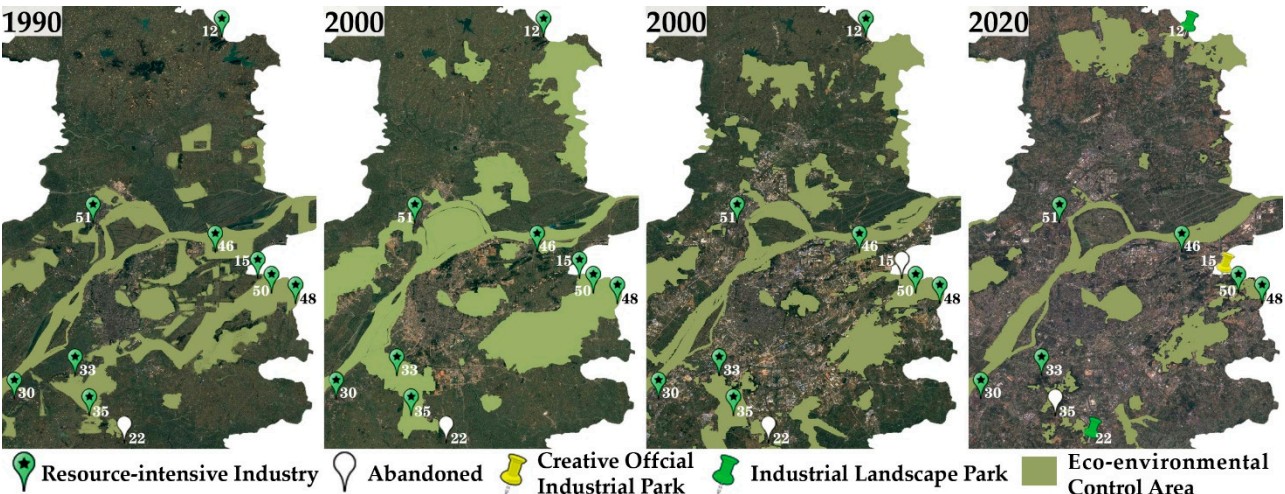

**Figure 27.** The change in the relationship between the operation state of resource-intensive industries and the eco-environmental control area over time (Figure source: own drawing; data source: on-site investigation and interview).

By accounting for the population density around Nanjing's existing resource-intensive industrial heritage sites, the average population density index of this group over four years was calculated. Figure 28 demonstrates that between 1990 and 2000, there was no obvious change in the operating quantity of resource-intensive industrial heritage sites and the population density around it. From 2000 to 2020, the number of abandoned and reused resource-intensive industrial heritage sites increased gradually, while the population density around them also showed the same trend. This indicates that the operation state of resource-intensive industrial heritage sites in Nanjing is directly affected by population distribution.

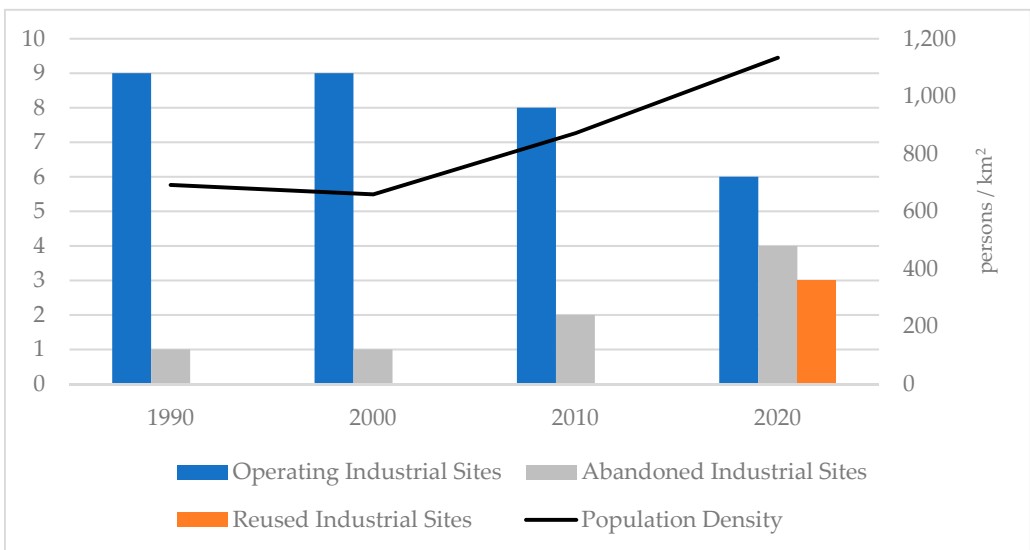

**Figure 28.** The change in the relationship between the operation state of resource-intensive industries and the population density over time (Figure source: own drawing; data from: [84]).

Figure 29 demonstrates that in the period from 1990 to 2000, no obvious change in the amount of investment funds for industrial renovation in Nanjing or the operation state of resource-intensive industrial heritage sites occurred. Between 2000 and 2010, the amount of investment funds for industrial renovation increased rapidly, while the operation state of resource-intensive industrial heritage sites changed slightly. However, the amount of investment funds for industrial renovation in the whole city has decreased significantly in the last ten years, with 33% of resource-intensive industrial heritage sites being shut down or reused. This shows that the amount of investment funds for industrial renovation may have a relatively small impact on the operation state of resource-intensive industrial heritage sites.

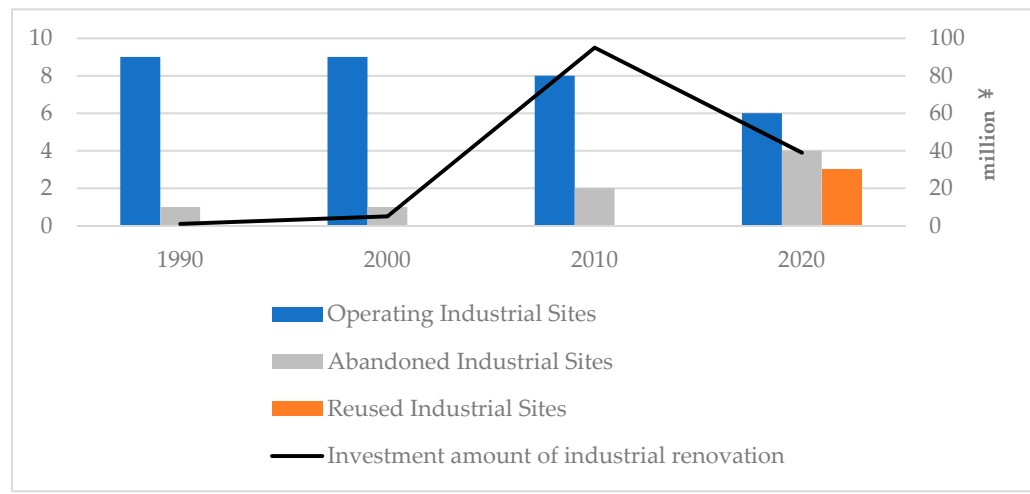

**Figure 29.** The change in the relationship between the operation state of resource-intensive industries and the investment amount of industrial renovation over time (Figure source: own drawing; data from: [83]).

By on-site interviews, the cases of abandonment and transformation of resource-intensive industrial heritage sites caused by the change in natural resources were counted. Figure 30 shows that 67% of the cases of abandonment or reuse were related to the deterioration of natural resources. The degree of natural resources deterioration had a great impact on these sites' operation states.

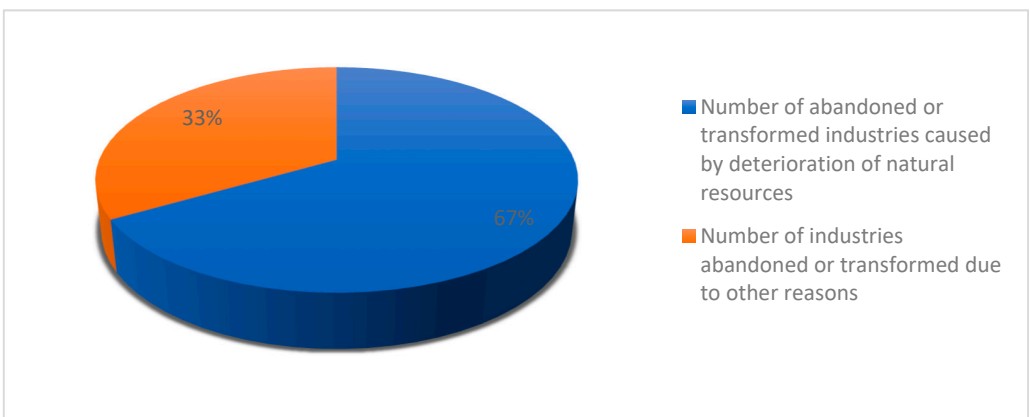

**Figure 30.** The relationship between the operation state of resource-intensive industries and natural resources deterioration (Figure source: own drawing; data source: on-site interviews).

*3.5. Reuse Characteristics of Nanjing Industrial Heritage Sites*

Figure 31 shows that over the past 30 years, the number of industrial heritage sites in operation in Nanjing has gradually declined, while the number of abandoned and reused industrial sites has gradually increased.

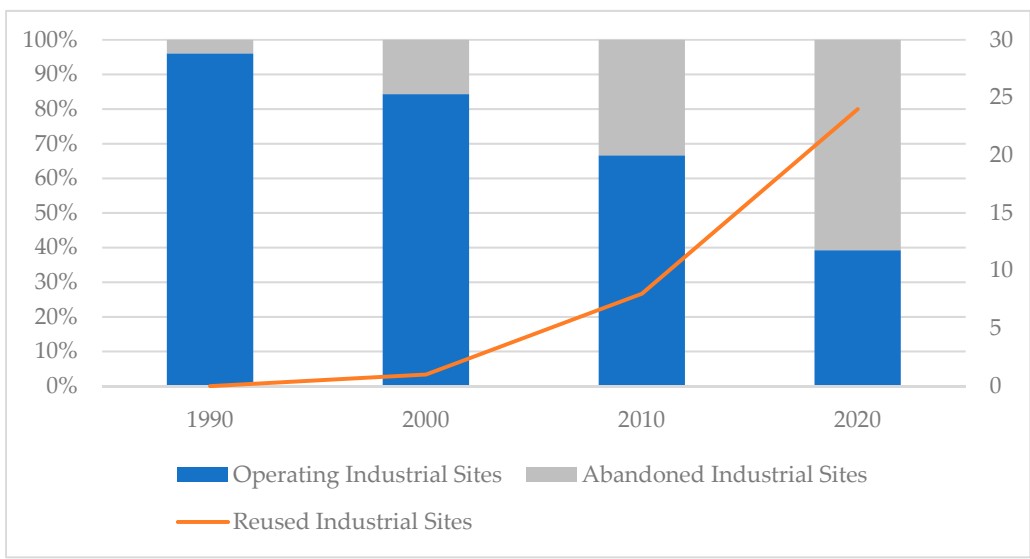

**Figure 31.** The change in the operation state of 51 industrial heritage sites in Nanjing over time (Figure source: own drawing; data source: on-site investigation).

From Figure 32a, we can see that Nanjing's modern industrial buildings are diverse in type and environmental conditions. The spatial layout of these heritage sites shows two typologies: a linear distribution along the river and a multi-center distribution in the city. The shape of the overall layout is stretched. Among the existing industrial heritage sites, capital-intensive industries account for a large proportion, which occupy an important position in Nanjing's economic structure and are located in the central area of the city.

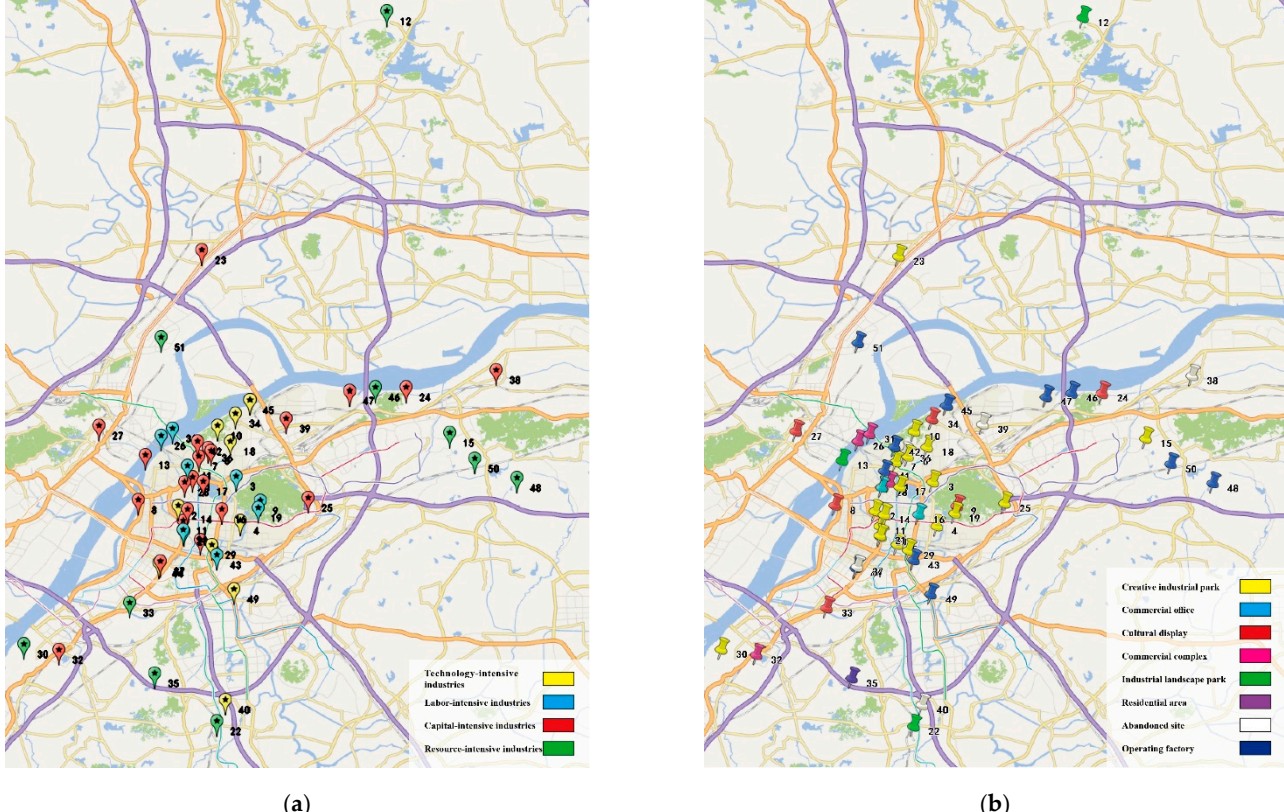

**Figure 32.** The original industry types and reuse status of the 51 industrial heritage sites in Nanjing: (**a**) distribution of industrial types of 51 industrial heritage sites in Nanjing; (**b**) reuse status of 51 industrial heritage sites in Nanjing (Figure source: own drawing; data source: on-site investigation and interview).

In Figure 32b, we can see that most of the existing industrial heritage sites in the urban central area have been reused, which to some extent reflects the government's policy of relocating industrial areas to the suburbs.

In addition, the practices of industrial heritage reuse in Nanjing have the following characteristics:

- The reuse purposes of most historical industrial sites were determined according to the characteristics of the geographical environment;
- The purpose of reuse varied, and most industrial heritage sites were reused in a varied manner, paying attention to the sustainability of building functions (Figure 33).

Due to the low public awareness of industrial heritage protection, the following problems also appeared in Nanjing's reuse cases:

- Figure 33 summarizes the excellent works of industrial heritage reuse in Nanjing. These cases pay more attention to the sustainability of a building's function, but they lack attention to landscapes remaining around the site, such as industrial communities spontaneously built around these sites.
- In the process of reusing industrial heritage sites, the authenticity and integrity of many industrial architectural heritage sites are lacking, for example, Nanjing Line Equipment Factory and Baijingyu Ophthalmic Pharmaceutical Factory (Figure 34).

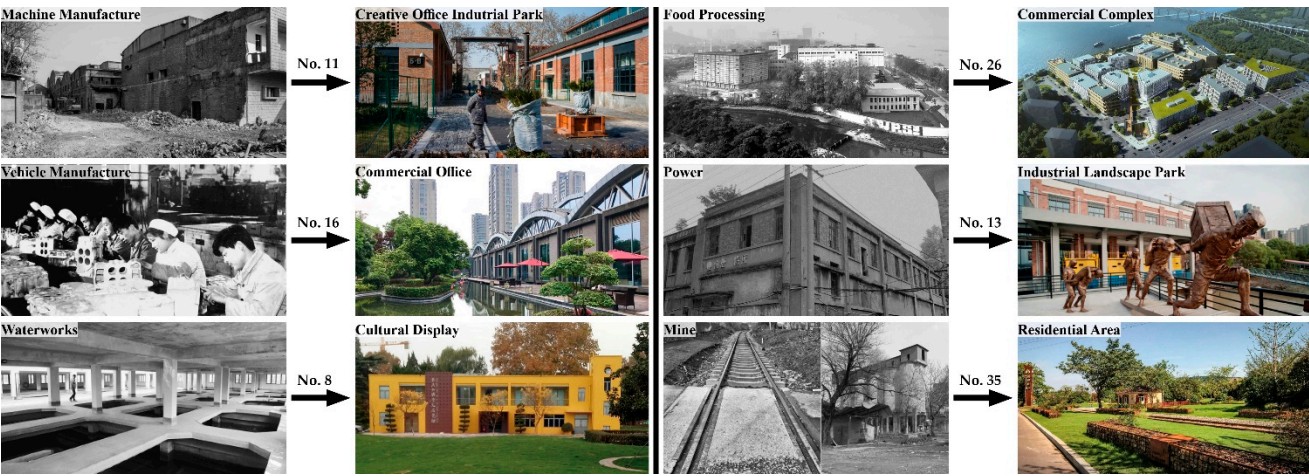

**Figure 33.** Typical cases of six reuse purposes of industrial heritage sites in Nanjing (Figure source: Zhou Qi Studio and on-site investigation), (The numbers in this figure represent the locations of these industrial sites in the maps above).

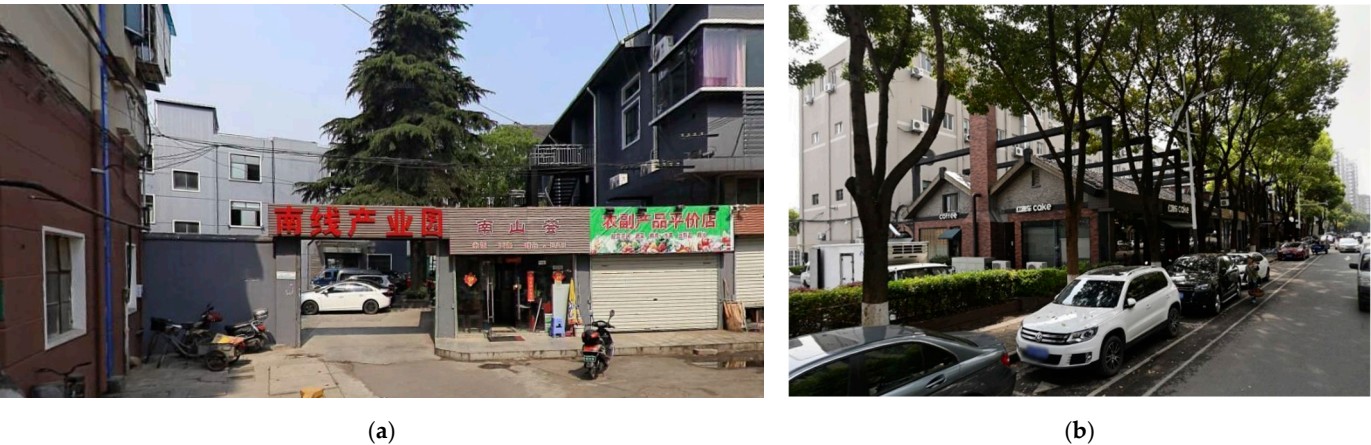

(**a**)                                    (**b**)

**Figure 34.** (**a**) Nanjing Line Equipment Factory; (**b**) Baijingyu Ophthalmic Pharmaceutical Factory (Source: photo by the author).

## 4. Discussion

In order to better understand the comprehensive influencing mechanism of the above six factors on the processes of industrial heritage site abandonment and regeneration in Nanjing, a simple analysis model is needed. This model, based on the quantitative analysis data of six influencing factors in paragraph 3, should clearly exhibit the influence of the six factors on different industrial types in Nanjing. We hope that through the intuitive expression of the model, protection and reuse strategies applicable to different categories of industrial heritage sites in Nanjing on an urban scale can be quickly identified. The specific composition of the model is shown in Figure 35. In order to qualitatively analyze the degree of influence of different factors on the operation state of Nanjing industrial heritage sites, we used specific scores for grading: "direct influence" = 100; "considerable influence" = 70; "relatively small influence" = 30; "little influence" = 10; "irrelevant" = 0.

According to the quantitative analysis data in paragraph 3, the degree of each factor's influence on the four different industry categories in Nanjing was scored. After statistics and induction, the influencing mechanism model was constructed (Figure 35), which can briefly and clearly show how the six factors affect the processes of industrial heritage site abandonment and regeneration in Nanjing.

Figure 35 indicates that to a certain extent, the urban growth boundary, traffic accessibility, eco-environment policy, population distribution, industrial renovation investment,

and natural resource changes all affect the abandonment and regeneration processes of industrial heritage sites in Nanjing. In addition, the degrees of influence of each factor on different categories of industrial heritage sites are obviously different.

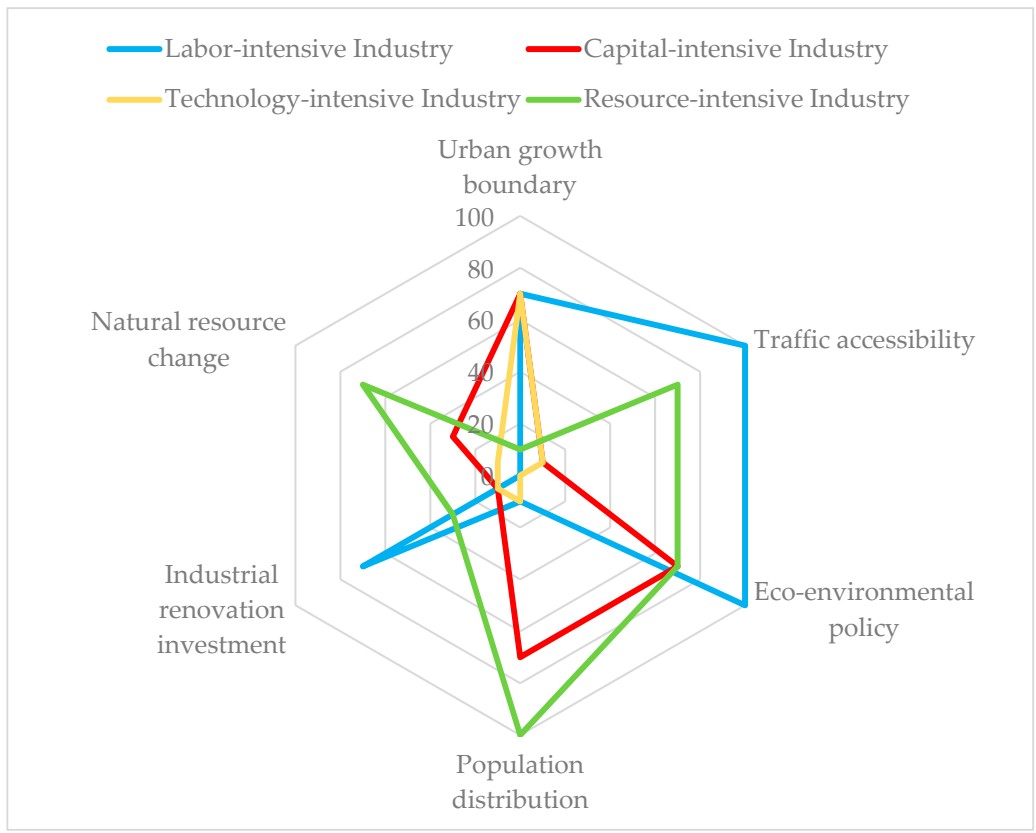

**Figure 35.** Comprehensive influence mechanism of the six factors on the operation state of Nanjing industrial heritage sites (Figure source: own drawing).

Specifically, the operation state of labor-intensive industrial heritage sites in Nanjing is mainly affected by the UGB and industrial renovation investment, especially traffic accessibility and eco-environmental policy. Since the new districts developed from early expansion cannot quickly form a perfect comprehensive service function, the historical urban area of Nanjing bears more pressure from living services, transportation and eco-environmental protection [13]. The economic structure in the city center gradually transforms from industry to service. In this context, labor-intensive enterprises, which usually have a small production scale, relatively low industrial status and less renovation funds, are more likely to be reused as service industries. This result also accords with Gaoli's research conclusion [99]. In a similar study, Min used a GIS spatial analysis technique and a Poisson regression model to analyze the mechanism of institutional impact on the spatial pattern of the manufacturing industry in the Wuhan metropolitan area since the 1990s. This research found that the convenient transportation in the city can reduce the transportation time and economic cost of enterprises, and it can promote the agglomeration of industrial enterprises, for instance, the greatest attraction of the expressway to labor-intensive industries [100]. With the increasingly efficient expressway system in Nanjing suburbs, labor-intensive industrial enterprises are more likely to relocate. By analyzing why other factors have little impact, we found that the original industrial types of the existing labor-intensive industrial heritage in Nanjing are light manufacturing, food processing, textiles, instruments and shipbuilding, which consume low amounts of natural resources for production. However, unexpectedly, the operation states of those sites that need large amounts of labor have little to do with population changes, which is consistent with the research result of Sang Jo [101].

The reason for this could be that quite a few residents lived in the historical area and already met the labor demand of those industrial sites long ago.

The UGB, eco-environmental policy and population distribution affect the operation state of Nanjing's capital-intensive industrial heritage sites greatly, especially the cultural heritage relics in old town. The capital-intensive industrial sites near the cultural heritage relics cannot exist too long, which indicates a good public awareness of cultural heritage protection. There are many capital-intensive industrial heritage sites in Nanjing, most of which are located in the historical area and pose a certain threat to the nearby eco-environment. The historical area can no longer meet the contemporary demand of Nanjing's high urbanization, and the original capital-intensive industries have to make concessions for the demand of Nanjing's economic structure adjustment and eco-environment preservation. This result is also consistent with the research conducted by Min [100]. In addition, with population growth and migration, the capital-intensive industries originally located in the suburbs of Nanjing gradually approached the newly built residential areas and were forced to shut down or relocate. In a similar study, Sang Jo also found that the population change in the new town was synchronized with the transformation of the industrial structure in this area [101]. Additionally, Min and Ling both found that transportation accessibility can promote the development of capital-intensive industries [100,102]. However, the results show that traffic accessibility has no obvious influence on the operation state of capital-intensive industrial heritage sites in Nanjing. This is because most of them are located beside the arterial roads in the old town with convenient traffic conditions. By analyzing why natural resources and industrial renovation investment have little impact on the operation of capital-intensive industrial heritage sites, we found that the production structure and abundant production equipment remaining from the early stage may be the answer.

An unexpected situation is that only the UGB led to the abandonment and reuse of technology-intensive industrial heritage sites. The UGB's influence on this type of industry should be related to the economic restructuring policy promoted by the Nanjing government. However, in the research of Min, the red line of the eco-environment has the most obvious influence on the location choice of technology-intensive enterprises, which have a high awareness of eco-environment protection [100]. Looking back at our analysis results, most technology-intensive industrial heritage sites in Nanjing are not in the eco-environment control area and thus are not affected by the ecological protection policy. By analyzing why other factors have little influence on technology-intensive industrial heritage sites, we found that their traffic conditions, production structure and production factor type are the reasons behind this phenomenon.

The operation state of resource-intensive industrial heritage sites is mainly affected by traffic accessibility, eco-environmental policy and changes in natural resources, especially population distribution. Convenient transportation usually means lower transportation costs and more social supervision for resource-intensive industries, which can effectively guarantee a good operation state and the sustainable development of the resource-intensive industry. Since all the existing resource-intensive industrial heritage sites in Nanjing are in the mining industry, which can affect the eco-environment and natural resources directly, this affects their own operation state. Additionally, with population growth and migration, the resource-intensive industrial heritage sites originally located in the suburbs of Nanjing are gradually approaching the newly built residential areas. The noise and air pollution of factories have a great impact on new residents, resulting in a large number of original resource-intensive industries being abandoned or reused. Finally, by analyzing why other factors have little influence on the operation of resource-intensive industrial heritage sites, we found the following results: (1) Nanjing's resource-intensive industrial heritage sites are isolated in the mountains, and the expansion of the urban area has little impact on them, which is also the reason that our result is not consistent with that of Gaoli [99]; (2) natural resources account for a large proportion of the production structure of resource-

intensive industry, so the investment funds for industrial renovation have little impact on its production activities.

Through the above analysis and discussion, on the urban scale, we can gain a macroscopic understanding of the processes of industrial heritage abandonment and regeneration in Nanjing. In order to answer the main research questions of this article, we need to further discuss whether we can design suitable planning strategies on this basis to deal with the specific issues of China's post-industrial reuse proposed in the first chapter.

As the six factors proposed in this study have different degrees of influence on the abandonment and regeneration processes of different categories of industrial heritage, the work of urban planning should pay more attention to the influence of planning policies on the reuse practice of industrial heritage and rationally account for all kinds of factors to guide and control this process.

The rapid urban expansion in China in the last 30 years has helped local governments to sell land on a large scale to obtain land finance and rapidly promote the local economy [103]. This met the needs of China's national conditions at that time, but it inevitably led to a large number of important industrial heritage sites being abandoned or reused, especially the labor-intensive and technology-intensive industrial enterprises in the old towns. In order for these industrial heritage sites to be properly and quickly protected in the process of urbanization, we suggest that the important labor-intensive industrial enterprises in the old towns should appropriately increase their industrial renovation investment funds to maintain their operations and start the emergency protection mechanism for abandoned labor-intensive industrial heritage sites.

Capital-intensive industrial heritage, as an important economic support of the city, should be carefully weighed in terms of their protection and reuse policies. Some polluting capital-intensive industrial enterprises should be shut down within a time limit and given sufficient time to relocate so as to minimize the negative impact on the eco-environment and urban economy. At the same time, delineating new residential areas should avoid important capital-intensive industries so as to avoid forced closure or relocation. The industrial enterprises that have caused interference to residential areas should be gradually relocated or reused on a small scale. The existing capital-intensive industrial heritage sites in the old towns generally occupy a large area. Therefore, in order to reduce their negative impact on the agglomeration effect of urban functions, reuse design work should reasonably transform the internal traffic system, so as to effectively connect it to the urban traffic system. Similar reference cases include the Zollverein site in Germany, where internal factories have been completely integrated with the surrounding residential areas through a reasonable road design, forming a complete, efficient industrial city [104].

Resource-intensive industries usually have a great impact on the environment due to their production structure. In order to ease the tension between ecological protection and economic development, the government should temporarily stop the production activities of resource-intensive enterprises that have caused damage to the eco-environment and restore their production qualifications after they adopt more eco-friendly production methods or transform into service industries. In addition, the inevitable noise problem in industrial production can be solved by planting greenery to isolate industrial production activities from the surrounding residents.

Based on the understanding of the processes of industrial heritage abandonment and regeneration in Nanjing, these planning strategies comprehensively consider multidisciplinary issues, such as ecological protection, transportation, population distribution, natural resources, and economy, in the post-industrial era. Through these proposals, urban planning departments will balance industrial heritage protection and urban development more systematically from a macro perspective. However, these planning strategies are general, and the urban problems in the post-industrial era are complex and numerous, e.g., How can the urban landscape be better shaped while protecting the value of industrial heritage? How can the function agglomeration effect of the city be improved without dismantling the industrial heritage? How can the ethical structure of a historical city be

changed by the production activity of industry? These issues require deeper analysis and more specific solutions in future research. In addition, some questions about industrial heritage reuse practice, such as the conflicts between stakeholders mentioned in the first chapter, the problems of industrial communities and social equality, the loss of authenticity and integrity of industrial heritage, should be better answered on a block or building scale.

## 5. Conclusions

This study expands the research perspective of industrial heritage reuse. Combined with time and space analysis, research on the abandonment and regeneration history of Nanjing industrial heritage sites in the past 30 years and its influencing factors at the macro level has been promoted. In contrast to the existing research on industrial heritage reuse, this paper analyzed the influencing factors of modern urban development that are closely related to industrial heritage reuse on the urban scale and fully considered urban society, politics, ecology, development environment and other aspects. In addition, a simplified analysis method of complex city dynamics based on official statistical data, historical maps and GIS technology was executed, aiming at providing a knowledge base and designing a clearer systematic planning strategy for the future development of existing industries. Through this work, the ability of urban planning departments to pre-control and guide industrial heritage reuse practice can be enhanced.

The influencing factors of industrial heritage reuse practice are multifaceted and complicated. An appropriate legislative and policy system is the precondition and first step for the effective development of reuse practice. This system should include a range of legal and institutional tools that can be used to regulate and control the conservation of industrial heritage. In this study, we proposed some strategies, which can help the Nanjing government improve its current legislative and policy system on the urban scale. However, this study has some limitations: First, a city is a huge and complex system, and the six influencing factors proposed in this study cannot fully reflect the influence of urban development on industrial heritage reuse practice. At present, some data in the Nanjing government's archives are not available or specific enough, thus some influencing factors can't be analyzed, e.g. The proportion of industrial land in the city and the density of urban buildings, which may also affect the reuse process. Second, the urban problems in the post-industrial era are complex and numerous, but the planning strategies proposed in this paper are not particularly in-depth. Detailed solutions will require more in-depth research on specific issues in the future. Finally, this study mainly analyzed the influencing factors of industrial heritage reuse practice in big cities at the macro level. However, the problems faced by industrial heritage reuse practice are not limited to the urban scale: they also significantly affect the community and building scales.

**Author Contributions:** Conceptualization, Y.W. and U.P.; methodology, Y.W.; software, Y.W.; validation, U.P., W.Q. and Q.Z.; formal analysis, Y.W.; investigation, Y.W.; resources, Y.W.; data curation, Y.W.; writing—original draft preparation, Y.W.; writing—review and editing, Y.W., U.P. and W.Q.; visualization, Y.W.; supervision, U.P., W.Q. and Q.Z.; project administration, U.P. and W.Q.; funding acquisition, Y.W. All authors have read and agreed to the published version of the manuscript.

**Funding:** This project is supported by the Chinese Scholarship Council, and the APC was funded by TU Delft Library.

**Data Availability Statement:** The data presented in this study were carried out as part of an ongoing Ph.D. research project at Southeast University and Delft University of Technology. It was conducted under the responsibility of Yanming Wu and supervision Uta Pottgiesser and Wido Quist.

**Acknowledgments:** This research is part of an ongoing Ph.D. research project.

**Conflicts of Interest:** The authors declare no conflict of interest. The funders had no role in the design of the study; in the collection, analyses, or interpretation of data; in the writing of the manuscript, or in the decision to publish the results.

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
