# Peer review of "The Guidance and Control of Urban Planning for Reuse of Industrial Heritage: A Study of Nanjing"

_land, doi:10.3390/land11060852_

Round 1
Reviewer 1 Report
This manuscript aims to provide an operational strategy for protecting and reusing industrial heritage in China, using Nanjing as a case study. Due to the ongoing debate on industrial heritage management in China, the manuscript's contribution to state of the art and literature is significant. As the authors try to develop a clearer, more systematic plan for the future of industrial heritage, more changes need to be made before it can be suggested for publication.
The concept of industrial heritage: the manuscript needs to evolve its definition of industrial heritage in the context of China. What is the role of the Wuxi Proposal compared with the international definition of industrial heritage (e.g., the Nizhny Tagil Charter)? Does its definition of industrial heritage resemble the international definition, except by distinguishing the historical periods of principal interest starting from the First Opium War (1840–1842) rather than the Industrial Revolution? [https://doi.org/10.1080/13527258.2019.1666293].
The Heritage value assessment: In the European context, it can be seen that there is an evolutional process from industrial archaeology to industrial heritage and now to the industrial landscape. Generally speaking, industrial heritage has gone from a specific interest in the monument (the individual building or a single machine) to the industrial sites (including the machines, buildings, and their infrastructure), then to the whole industrial area and industrial landscape. This process needs to be briefly highlighted in the literature review. However, please provide more in-depth information about the specific values and features used to define industrial heritage in the Chinese context compared to the European context. [https://doi.org/10.6092/issn.1973-9494/9225]
Does the reuse of industrial buildings and sites in Nanjing have to do with the goal of making the city more economically prosperous?
The Industrial heritage reuse strategy: China's industrial heritage may have a shift towards multiple forms of reuse, either being frozen as museums or "self-saving" reuses or the rehabilitation of abandoned industrial buildings by stakeholders. For the second strategy, the protection of industrial heritage needs to maintain a balance between retaining the intrinsic value of the past and meeting new demands. It would be proper to criticize the Nanjing reuse practice in terms of these two concepts and eventually develop a conceptual framework that helps understand the public’s demands and suggests the way industrial heritage should be protected and reused. I also want to know how the Nanjing experience can help ease the tension between conservation and regeneration.
[https://doi.org/10.3390/land11010016].
Please note that developing an appropriate legislative and policy system is the first step to reusing industrial heritage in Chinese cities. It should be clarified in your manuscript. This system should include a range of legal and institutional tools that can be used to regulate and control the conservation of industrial heritage.
Good Luck
Reviewer 2 Report
The reuse of industrial heritage is a new concept, which is an extension of the ancient cultural heritage. The point is that this post-industrial activity in order to be sustainable must not appear as an independent development in a nation’s history but a follow up. A combined land use for post industrial cultural reuse entangled to earlier cultural periods , but present in same land use, project, is necessary to mention. Independent reuses are not a concept, instead a unified concept is integrated and most welcome for social educational research innovation purposes. Specific post-industrial reuse issues must be mentioned in specific and the capital-intensive industries and the eco-environmental control area changing with time should take into account cultural heritage relics and how are these unified into the project. The readers of LAND would benefit more to see integration of the presented excellent work with some issues I recall and appear a more unified and balanced work with an international perspective.
- Thus I would like to see the abovementioned coupled with proper references e.g.
Serap Ünal (2021) ceramic production from neolithic doğanhisar pottery culture: intangible heritage of technology transfer from past to present, Scientific Culture, Vol. 7, No. 3, pp. 77-91 DOI: 10.5281/zenodo.5062884
Regarding cultural tourism;
Brychan Thomas, Simon Thomas and Lisa Powell (2017)the development of key characteristics of welsh island cultural identity and sustainable tourism in wales, SCIENTIFIC CULTURE, Vol. 3, No. 1, (2017), pp. 23-39 DOI: 10.5281/zenodo.192842
Bushra Obeidat and Hamzeh Miqdady (2021) preserving the traditional watermills in wadi orjan jordan using an analytic hierarchy process, SCIENTIFIC CULTURE, Vol. 7, No. 1, pp. 21-30 DOI: 10.5281/zenodo.4107177
Regarding conservation/reuse and management very important factors more updated works is recommended:
Hassan S.A. Mahmoud (2021) multiscientific approach for the characterization and assessment of the degradation state of the historical al-shafi’i mosque walls (jeddah, kingdom of saudi arabia). SCIENTIFIC CULTURE, Vol. 7, No. 1, (2021), pp. 1-19 DOI: 10.5281/zenodo.4107161
Moreno, M., Ortiz, P. and Ortiz, R. (2019) VULNERABILITY STUDY OF EARTH WALLS IN URBAN FORTIFICATIONS USING CAUSE-EFFECT MATRIXES AND GIS: THE CASE OF SEVILLE, CARMONA AND ESTEPA DEFENSIVE FENCES. Mediterranean Archaeology and Archaeometry Vol. 19, No 3, 119-138 DOI: 10.5281/zenodo.3583063
Raed Alghazawi, Mohammed Waheeb, Dima Kraishan (2015) CULTURAL HERITAGE MANAGEMENT ADAPTATION IN THE ARAB WORLD: REVIEW & PERSPECTIVES, Mediterranean Archaeology and Archaeometry, Vol. 15, No 1, 11-21
Abdelhalim Assassi and Ammar MebarkiSPATIAL (2021) Spatial configuration analysis via digital tools of the archeological roman town Timgad, algeria, Mediterranean Archaeology and Archaeometry Vol. 21, No 1,71-84. DOI: 10.5281/zenodo.4284429
Evenmore the reuse project must be harmonized with ethical principles and moreover draw elements from the urban planning and use of those products. See for example:
Baya Belmessaoud Boukhalfa (2022) Space syntax theory identifies the ethical reversal trend of the overwhelmed mādina of al-djaza’ir urban morphology, Mediterranean Archaeology and Archaeometry Vol. 22, No 1, pp. 155-182 DOI: 10.5281/zenodo.6464956
And always following the principles of UNESCO
Naif A. Haddad , Leen A. Fakhoury and Yasir M. Sakr (2021) A critical anthology of international charters, conventions & principles on documentation of cultural heritage for conservation, monitoring & management, Mediterranean Archaeology and Archaeometry Vol. 21, No 1, 291-310 DOI: 10.5281/zenodo.4575718
Reviewer 3 Report
The treated article deserves a high qualification for the specialization of the subject.
The methodological approach responds to the minimum requirements usually established and the results obtained are of special interest to science.
The bibliography is extensive and sufficiently related.
A revision in grammatical forms is suggested.
I resolve to positively rate the article presented.
Reviewer 4 Report
- Line 31: Please define “CPC” and all other acronyms at first mention. The introduction should highlight the limitations of prior studies on industrial site redevelopment in China and underscore how the present study builds on them and fill their research gaps.
- There is also the need to provide clear theoretical relevance to the study. For example, the theory of sustainable cities links brownfield redevelopment to reducing sprawl, cost-saving n infrastructure provision in the city, and reducing commuting costs such as congestion and carbon emissions. The relevance of the study to SDG 11 should also be emphasized.
- Section 2.1. There is a need for a strong justification for selecting Nanjing as a study area.
- Section 2.2: Provide the reference (e.g., website) to data sources for urban growth boundary and Chart 1
- Pages 9: justify all the selected measures listed in bullets
- Lines 350-352: what are the likely factors responsible for the pattern of the reported results?
- Section 4: the discussion section should compare the findings with similar studies. Nevertheless, not a single piece of literature is cited in the section.
- The section should also indicate the extent to which the research questions are answered.
- Section 5: the conclusion section should be more concise. Figures and results are better placed in the discussion section. The conclusion should summarize the key lessons of the paper and its implications for urban development policy and practice.
Round 2
Reviewer 1 Report
The authors have replied to the suggestions and concerns raised by this reviewer in a very satisfying manner, and I can see that the revised manuscript has much improved. Thank you.
Reviewer 2 Report
well restructured.
This manuscript is a resubmission of an earlier submission. The following is a list of the peer review reports and author responses from that submission.